# Gut microbiome of the largest living rodent harbors unprecedented enzymatic systems to degrade plant polysaccharides

Lucelia Cabral[1,8], Gabriela F. Persinoti [1,8 ✉], Douglas A. A. Paixão[1,8], Marcele P. Martins[1,2], Mariana A. B. Morais [1], Mariana Chinaglia[1,2], Mariane N. Domingues [1], Mauricio L. Sforca[3], Renan A. S. Pirolla[1], Wesley C. Generoso [1], Clelton A. Santos [1], Lucas F. Maciel[1], Nicolas Terrapon [4,5], Vincent Lombard[4,5], Bernard Henrissat [6,7] & Mario T. Murakami [1 ✉]

The largest living rodent, capybara, can efficiently depolymerize and utilize lignocellulosic biomass through microbial symbiotic mechanisms yet elusive. Herein, we elucidate the microbial community composition, enzymatic systems and metabolic pathways involved in the conversion of dietary fibers into short-chain fatty acids, a main energy source for the host. In this microbiota, the unconventional enzymatic machinery from Fibrobacteres seems to drive cellulose degradation, whereas a diverse set of carbohydrate-active enzymes from Bacteroidetes, organized in polysaccharide utilization loci, are accounted to tackle complex hemicelluloses typically found in gramineous and aquatic plants. Exploring the genetic potential of this community, we discover a glycoside hydrolase family of β-galactosidases (named as GH173), and a carbohydrate-binding module family (named as CBM89) involved in xylan binding that establishes an unprecedented three-dimensional fold among associated modules to carbohydrate-active enzymes. Together, these results demonstrate how the capybara gut microbiota orchestrates the depolymerization and utilization of plant fibers, representing an untapped reservoir of enzymatic mechanisms to overcome the lignocellulose recalcitrance, a central challenge toward a sustainable and bio-based economy.

[1] Brazilian Biorenewables National Laboratory, Brazilian Center for Research in Energy and Materials, Campinas, SP, Brazil. [2] Graduate Program in Functional and Molecular Biology, Institute of Biology, University of Campinas, Campinas, SP, Brazil. [3] Brazilian Biosciences National Laboratory, Brazilian Center for Research in Energy and Materials, Campinas, SP, Brazil. [4] The Institut National de la Recherche Agronomique, USC 1408 AFMB, 13288 Marseille, France. [5] Architecture et Fonction des Macromolécules Biologiques, CNRS, Aix-Marseille Université, Marseille, France. [6] Department of Biotechnology and Biomedicine (DTU Bioengineering), Technical University of Denmark, 2800 Kgs Lyngby, Denmark. [7] Department of Biological Sciences, King Abdulaziz University, Jeddah, Saudi Arabia. [8] These authors contributed equally: Lucelia Cabral, Gabriela F. Persinoti, Douglas A. A. Paixão. ✉email: gabriela.persinoti@lnbr.cnpem.br; mario.murakami@lnbr.cnpem.br

The symbiotic microbiota within the digestive tract of herbivores has been an overwhelming source of diverse enzymatic mechanisms for lignocellulose depolymerization[1–4]. For decades, the microbiota of foregut (rumen) fermenters has been employed as a model system[5,6], which resulted in the discovery of sophisticated systems to degrade recalcitrant plant fibers, such as the multi-enzyme complexes (cellulosomes) from *Ruminococcus flavefaciens*[7] and the efficient cellulose degradation system from *Fibrobacter succinogenes*[8].

A less explored and equally effective class of herbivores is the hindgut fermenters[9]. Similar to foregut fermenters, the digestion is accomplished by a symbiotic microbial community, but in a single fermentation chamber[10]. These monogastric herbivores comprise a vast range of animals, from massive mammals, such as elephants, rhinos, and horses, to small animals, as rabbits and semi-aquatic rodents[11]. In addition, they are spread over a myriad of ecological niches, suggesting that highly specialized specialized microbial strategies may have emerged to overcome the complexity and diversity of plant glycans in these environments.

The capybara (*Hydrochoerus hydrochaeris*) is the largest living rodent, typically found in the Pantanal wetlands and Amazon basin, and it is also known as "the master of the grasses" due to its diet based on gramineous and aquatic plants. In this animal, the fermentation takes place in the cecum that corresponds to almost three-quarters of the gastrointestinal tract, reaching a digestive efficiency comparable to that of ruminants[12]. Moreover, as a strategy to maximize absorption of nutrients derived from bacterial fermentation, capybara can eat their cecotropes, a specific type of soft excreta[13]. This habit of cecotrophy is more frequent in wild animals during the dry season, when food is scarce[13]. Despite being a formidable plant biomass fermenter, the enzymatic strategies and metabolic pathways employed by its microbial symbiotic community for the breakdown and utilization of recalcitrant dietary fibers remain mostly elusive. In addition, wild capybara animals dwelling the Southeast region of Brazil have incorporated sugarcane in their diet for decades[14], which makes their cecal microbiome particularly attractive for the discovery of enzymatic mechanisms for the depolymerization of this industrially relevant feedstock and related grasses.

To elucidate the enzymatic strategies employed by the Brazilian capybara microbiota for plant cell wall depolymerization, we comprehensively investigated this gut microbial community combining 16S rRNA gene targeting sequencing (16S), metagenomics (MG), metatranscriptomics (MT), and nuclear magnetic resonance (NMR) based metabolomics, with carbohydrate enzymology and X-ray crystallography, which ultimately led to the discovery of two families, according to the carbohydrate-active enzymes database (CAZy) (Fig. 1). These findings highlight the potential of the capybara gut microbiome as a reservoir of uncharted enzymatic systems for carbohydrate processing, and thereby expanding our current understanding of gut microbial strategies from hindgut fermenters to overcome the plant cell wall recalcitrance, which might be instrumental to foster the development of value-added products from lignocellulosic agro-industrial materials.

## Results

**Capybara gut microbiota encompasses taxonomic novelties involved in plant fiber breakdown.** The taxonomic structure of the capybara gut microbiota from fresh cecal and rectal samples is mainly constituted by Bacteria, with only a small fraction of reads corresponding to Archaea (16S: 1.07%, MG: 0.40%, and MT: 1.55%) and Fungi (MG 0.12% and MT: 0.32%). 16S rRNA gene-based taxonomic analysis, corroborated by 16S rRNA reads recovered from metagenome (16S_MG), MG, and MT data sets,

indicates that the most abundant bacteria found in this microbiota are members from the phyla Firmicutes (mean ± SD: 35.8 ± 12.4%) and Bacteroidetes (31.5 ± 9.8%), followed by Fusobacteria (15.3 ± 5.4%) and Proteobacteria (8.4 ± 5.4%) (Fig. 2a, b and Supplementary Fig. 1). Initial characterization by hybridization approaches of the capybara cecal microbiota from wild Venezuelan animals (not exposed to sugarcane biomass) also identified Firmicutes, Proteobacteria, and Bacteroidetes as major constituents of this microbiota[15]. However, distinct abundances of these phyla were observed with a higher number of OTUs belonging to Firmicutes (322) and Proteobacteria (301) compared to only 76 OTUs from Bacteroidetes[15]. This variability would be an indicative of a potential feed specialization such as from wild animals dwelling in the Brazilian sugarcane belt, and/or microbial dynamics and adaptation to nutrient availability. A prevalence of Firmicutes, Bacteroidetes, and Proteobacteria was also observed in the gut microbiota of beaver, horse, rabbit, and koala[16–20], indicating to be a common feature among hindgut herbivores. Comparative investigation of the gut microbiota of a large number of carnivores, omnivores, and herbivores, observed a lower microbiota biodiversity in cecum fermenters (such as capybara, beaver, and rabbit) compared to colon fermenters (such as horse, elephant, and tapir), despite their high plant fiber-based diets[21]. The combination of biomass adaptation/specialization and low biodiversity of cecum microbiota would suggest the presence of highly efficient enzymatic systems for the depolymerization and utilization of recalcitrant plant fibers. In this regard, the reconstruction of metagenome-assembled genomes (MAGs) (Supplementary Data 1) revealed several taxonomic novelties, representing either unknown species or genera from bacterial families that are recognized as plant fiber degraders such as Fibrobacteraceae and Bacteroidaceae (Fig. 2c). Among these taxonomic novelties, MAGs 41 to 44, assigned to the uncultured UBA932 family, are phylogenetically grouped and may represent an unprecedented Bacteroidetes genus with a genetic potential for lignocellulose degradation. Interestingly, the high-quality MAG42 harbors a predicted polysaccharide utilization locus (PUL), which culminated in the discovery of the GH173 family (see the "Biochemical" section below). MAGs 56 to 59 may also represent an expansion of genera in the Bacteroidaceae family and are closely

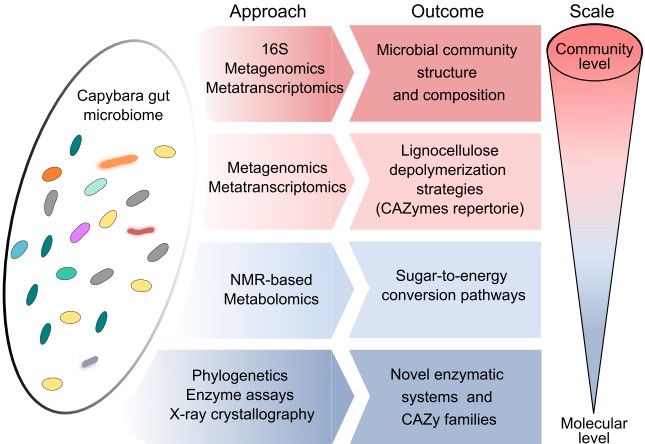

**Fig. 1 Experimental design employed to explore capybara gut microbiome.** An integrated multi-omics approach from the community to the molecular level, employing 16S rRNA gene targeting sequencing (16S), metagenomics, metatranscriptomics and nuclear magnetic resonance (NMR) based metabolomics along with phylogenetics, enzyme assays, and X-ray crystallography were used to investigate plant fiber depolymerization strategies and major energy conversion pathways exploited by capybara gut microbiota.

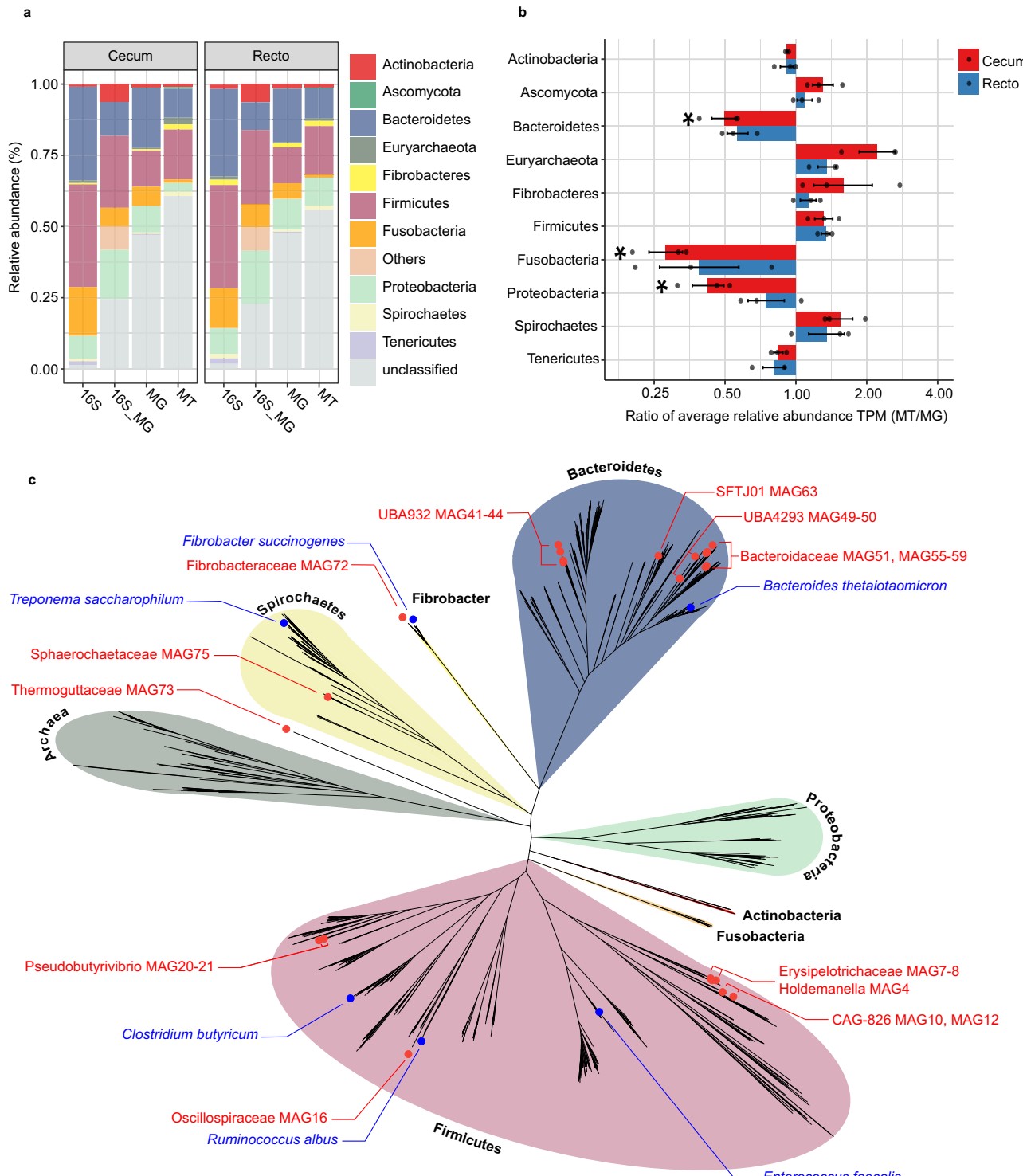

**Fig. 2 Microbial taxonomic composition of capybara gut microbiome. a** Relative phyla abundance based on 16S rRNA gene target sequencing (16S), 16S rRNA recovered from metagenome (16S_MG), whole metagenome (MG) and metatranscriptome (MT) reads. **b** Ratio of average relative phyla abundance (Transcripts Per Million - TPM) of metatranscriptome to metagenome (MT/MG), expressed in a $\log_2$ scale. Bars marked with an asterisk (*) are significantly different from MT/MG = 1 (two-tailed *t*-test, *P*-value with false discovery rate correction for multiple tests, 0.05 was used as the threshold of significance level). The data represent the average of three independent experiments (*n* = 3), using fresh samples collected from the cecum and recto of three wild animals by abdominal surgical procedure. **c** Phylogenetic tree of the 79 recovered metagenome-assembled genomes (MAGs) from capybara microbiota, including related genomes publicly available. The capybara microbiota taxonomic novelties and some reference genomes are highlighted with red and blue circles, respectively. Source data are provided as a source data file.

related to other uncultured Bacteroidaceae MAGs recovered from sheep, elephant and mice gut microbiomes[22] (Supplementary Fig. 2). In particular, the high-quality MAG57 notably contains one of the largest inventories of genes encoding carbohydrate-active enzymes (CAZymes) among the 79 recovered MAGs from capybara gut microbiota (Supplementary Data 2), which are likely involved in the depolymerization of distinct hemicelluloses, including heteroxylans, an abundant polysaccharide in sugarcane and other grasses (see the "Biochemical and structural" section below). These analyses indicate that the capybara gut microbiota may harbor unexplored and high-performance molecular systems for plant fiber breakdown and utilization.

**Fibrobacteres and Bacteroidetes are main degraders of dietary fibers in capybara gut.** In order to understand the ability of capybara gut microbiota to convert plant polysaccharides into edible sugars, MG and MT data from cecal and rectal samples were investigated to determine the genomic potential associated with plant fiber depolymerization. A total of 7377 putative CAZymes genes encoding for 106 glycoside hydrolases (GH), 11 carbohydrate esterases (CE), and 10 polysaccharide lyases (PL) families were identified, of which 517 genes presented a modular architecture (Supplementary Data 3). The most abundant CAZymes identified are members of the families GH3, GH2, and GH1 (by decreasing abundance) (Fig. 3), which is in agreement with that reported for other gut microbiomes such as human, swine, and cattle rumen[23]. These enzymes encompass diversified activities such as β-glucosidase, β-xylosidase, β-galactosidase, β-mannosidase, and α-arabinofuranosidase, and are often associated with the final steps in the depolymerization cascade of several plant polysaccharides such as cellulose, heteroxylans, mixed-linkage β-glucans, and β-mannans.

In the CAZyme repertoire of this microbiota neither cellulases from families GH6, GH7, and GH48, nor cellulosomes, assessed by the presence of cohesin and dockerin domains associated with cellulases, could be identified. In ruminal *Ruminococcus*[24] and anaerobic fungi, these families are found in high abundance, possibly targeting recalcitrant cellulose structures[25]. However, in the capybara gut microbiota assayed here, fungi were detected only at very low abundance (Fig. 2a). It suggests that cellulose degradation in the capybara gut might be mainly accomplished by endo-β-1,4-glucanases (Enzyme Commission (EC) number 3.2.1.4) from families GH5 (subfamilies GH5_2 and GH5_4), GH8, GH9, and GH45, which were detected either as single domains or in multi-modular protein architectures (Supplementary Data 3). Interestingly, the most expressed genes encoding endo-β-1,4-glucanases belong to families GH5_2, GH8, GH9, and GH45, and were identified in *Fibrobacter* MAGs (Supplementary Fig. 3 and Supplementary Data 4) that also present a high MT/MG ratio (Fig. 2b), indicating that these bacteria might play a key role in cellulose degradation in the capybara gut. *Fibrobacter succinogenes* is known as a highly efficient cellulolytic bacterium in the cow rumen[26] and employs a multi-protein complex to attach to cellulose fibers and cellulases secreted by the T9SS-dependent secretion system for cellulose breakdown[27]. The three *Fibrobacter* MAGs recovered from capybara gut microbiome encode cellulases with a T9SS signal sequence as well as proteins for cellulose adhesion including tetratricopeptide, fibro-slime, OmpA, and pilin proteins, as reported for *F. succinogenes*[27]. Furthermore, from the set of 347 proteins observed in the outer membrane vesicles (OMVs) from *F. succinogenes*[28], we have identified 262 with sequence identity ranging from 30 to 99%. These observations suggest that *Fibrobacter* mechanisms, fundamentally relying on cell surface adhesion and OMVs, might be central for cellulose depolymerization in the capybara gut.

In the multiple recovered Bacteroidetes MAGs, a large number of PULs and clusters of CAZymes (CCs) were identified (Fig. 4 and Supplementary Data 5), which provide highly diversified capabilities to this microbiota to cope with the chemical and structural complexity of abundant hemicelluloses and pectins in gramineous or aquatic plants such as heteroxylans, mixed-linkage β-glucans, β-1,3-glucans, xyloglucans, mannans, and homogalacturonans. Notably, the identified PUL targeting mixed-linkage β-glucans (Supplementary Fig. 4) is conserved in capybara and human microbiomes, presenting identical gene architecture encompassing one GH16 and two GH3 enzymes[29]. PULs targeting heteroxylans and homogalacturonans (Supplementary Fig. 4), common components of gramineous plants such as sugarcane[30], also resemble PULs identified in human gut microbiomes[31], highlighting a remarkable level of conservation of Bacteroidetes enzymatic systems in omnivores and hindgut herbivores.

Despite the presence of multiple CEs (Fig. 4), the lack of auxiliary-active enzymes (AAs) indicates a low capacity of the capybara microbiota to perform plant biomass delignification, as also observed for other monogastric herbivores[32,33]. As a mechanism to cope with lignin-rich diets, these animals may employ cecotrophy to enhance digestibility and nutrient uptake. In addition, it is noteworthy that many identified PULs only showed similarity with non-experimentally validated PULs and without a defined substrate target (Supplementary Data 5), which in part could be due to intrinsic limitations of genome reconstruction from metagenomes, but could also reflect the variability, heterogeneity and limited knowledge of the structure and composition of the glycans present in the diet of wild capybaras.

In summary, the CAZyome (CAZyme inventory) analysis of the capybara gut microbiota indicates Fibrobacteres as the main drivers for cellulose breakdown, whereas the numerous Bacteroidetes PULs and clusters of CAZymes confer to this community a myriad of enzymatic strategies to tackle with complex and diverse hemicelluloses and pectins commonly present in gramineous and aquatic plants, major components of capybara diet.

**Metabolite profiling shows high performance on the conversion of dietary fibers into short-chain fatty acids.** Once addressed important players for depolymerization of dietary fibers in the capybara gut, we further investigated the role of these microorganisms in the conversion of free sugars into energy for the host by integrating metabolomics (obtained from the polar fraction after a Folch extraction) and metabolic reconstruction analysis.

The major fermentation products measured in the capybara gut were short-chain fatty acids (SCFAs), among more than 40 metabolites detected by NMR spectroscopy-based metabolomics (Supplementary Data 6). The most abundant metabolites observed in cecal and rectal samples were acetate (mean ± SD: 74.83 ± 22.17 and 30.40 ± 22.76 μM, respectively), propionate (31.0 ± 6.67 and 15.98 ± 12.8 μM), and butyrate (23.30 ± 5.63 and 8.35 ± 12.83 μM). These SCFA ratios indicate a forage-based diet and are similar to that seen for ruminants[34,35], supporting a high efficiency of this microbiota in the use of dietary fibers as an energy source.

Genes related to pyruvate fermentation into acetate were highly abundant in both MG and MT data for cecal and rectal samples, and they are derived from Firmicutes, Bacteroidetes, and Fusobacteria (Fig. 5 and Supplementary Fig. 5). Metabolic pathway reconstruction analysis shows that acetate can be produced by any of the bacterial MAGs recovered from capybara gut microbiome (Fig. 5), which is in agreement with the high

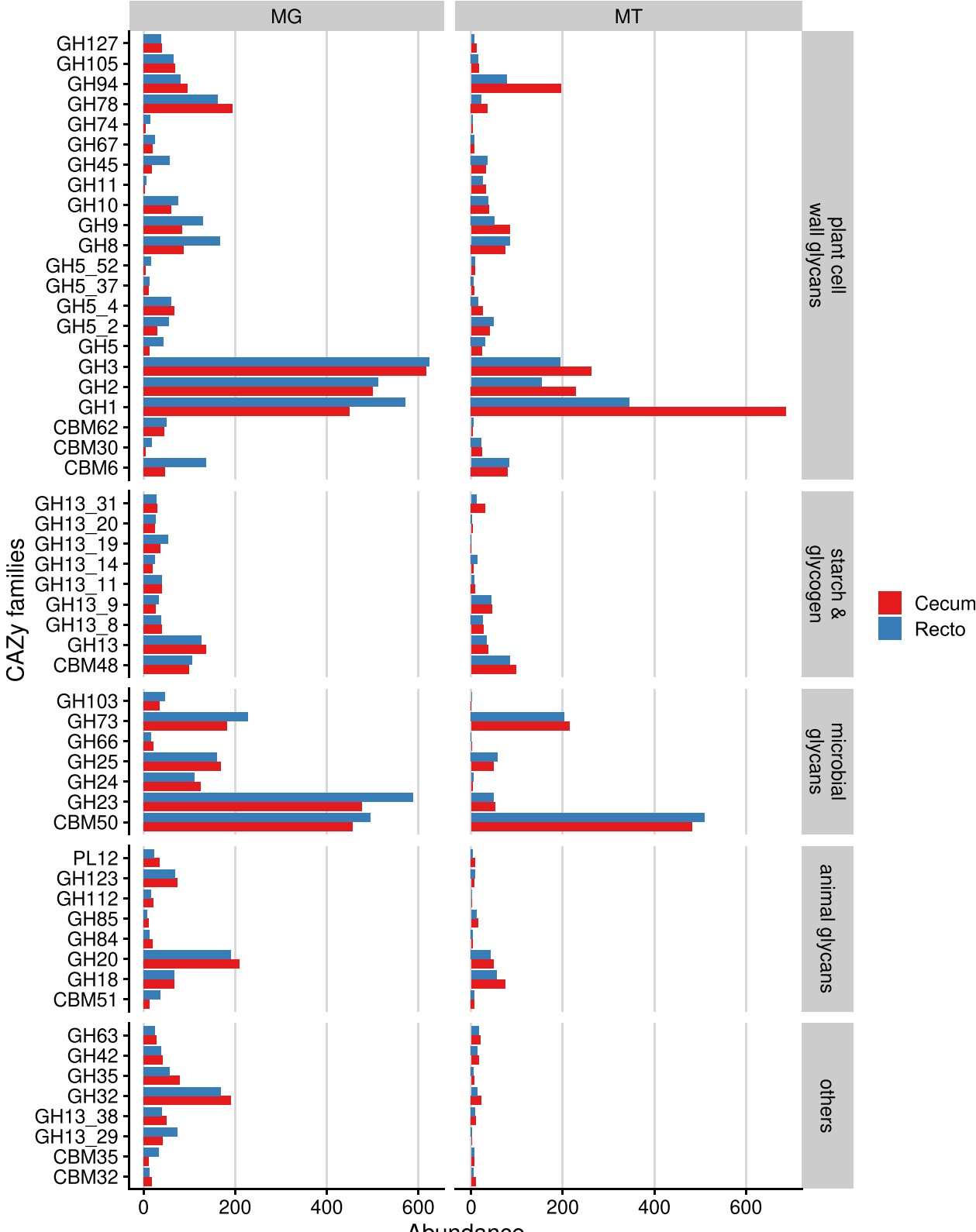

**Fig. 3 Functional annotation of capybara metagenome co-assembly predicted genes according to carbohydrate-active enzymes database (CAZy).** Cumulative abundance of main CAZy families predicted to act on distinct carbohydrates. Abundance is expressed as the cumulative TPM (Transcripts per Million). MG (metagenome) and MT (metatranscriptome) represent the average of three independent experiments ($n = 3$), using fresh samples collected from the cecum and recto of three wild animals by abdominal surgical procedure. Source data are provided as a source data file.

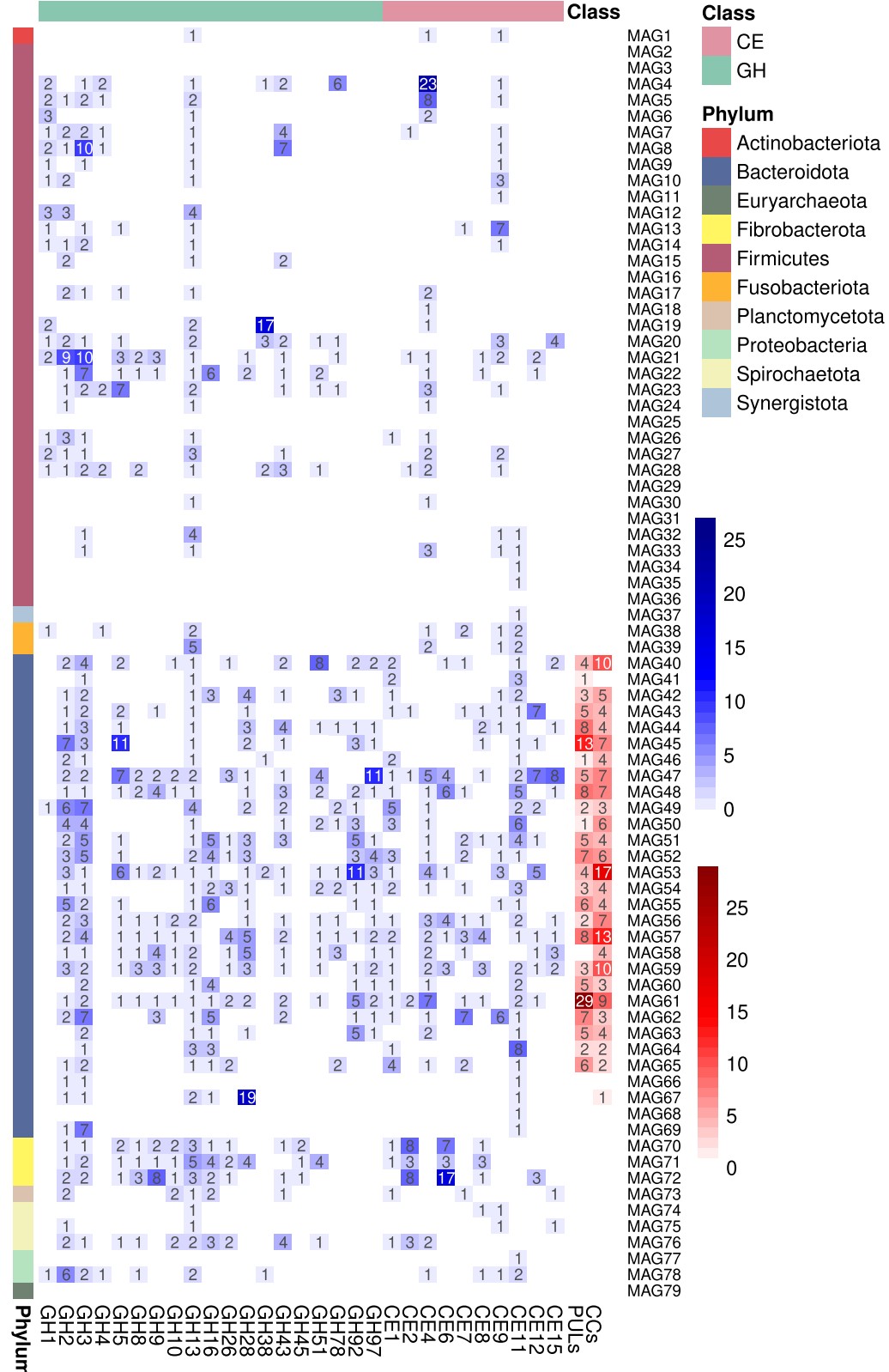

**Fig. 4 Heatmap of main carbohydrate-active enzymes (CAZymes) families, polysaccharide utilization loci (PULs) and clusters of CAZymes (CCs) identified in the recovered metagenome-assembled genomes (MAGs).** The heatmap indicates the number of genes encoding CAZymes, PULs, and CCs. The number of PULs and CCs are colored in a red scale, whereas the number of proteins in each family are in a blue scale.

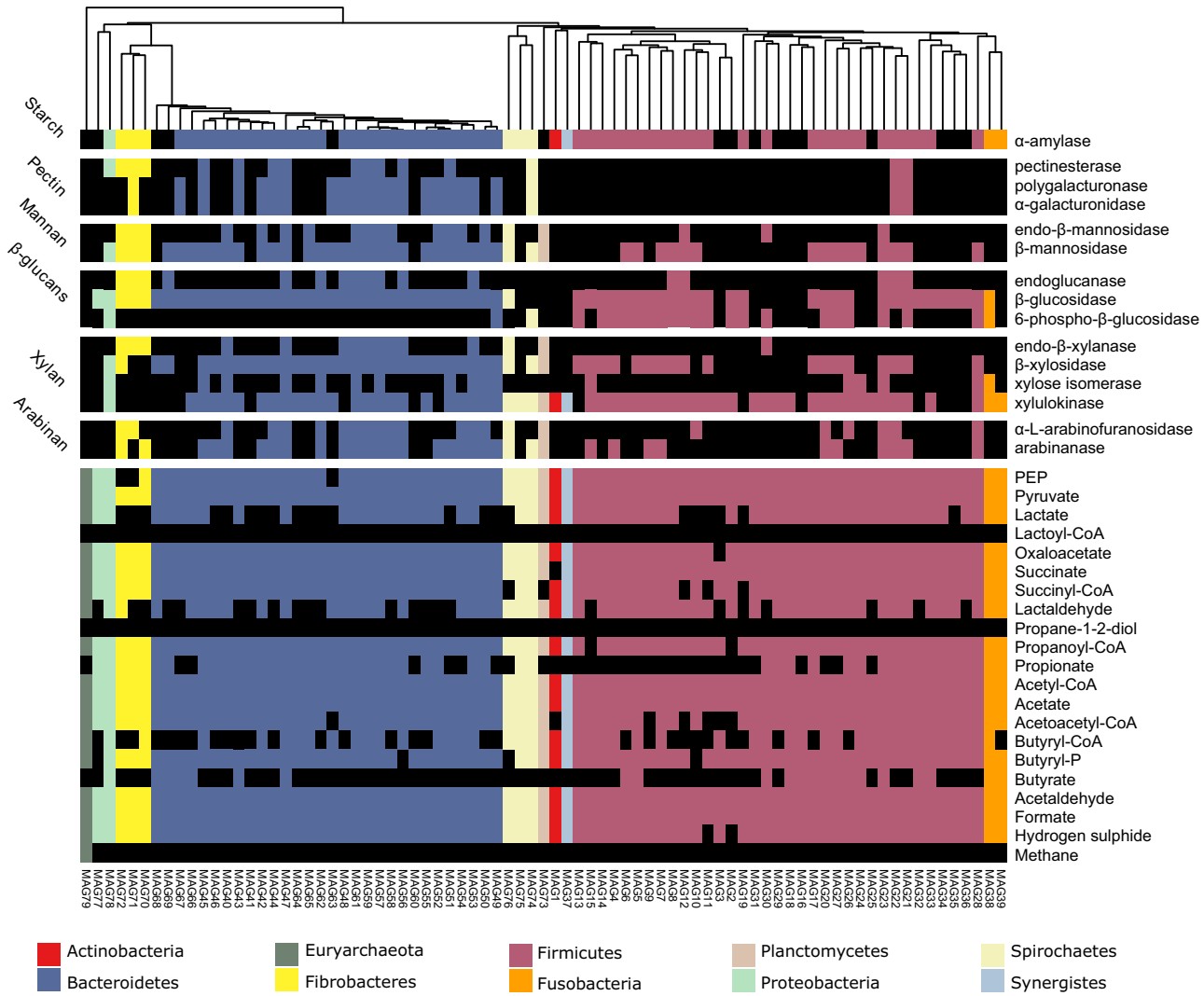

**Fig. 5 Metabolic reconstruction of 79 unique metagenome-assembled genomes (MAGs) recovered from the capybara gut microbiome.** Heatmap indicates the presence or absence of enzymes related to plant polysaccharides degradation or metabolites production (listed on the right) in each MAG (bottom) according to their set of genes. The presence of each enzyme/metabolite is denoted by a box colored based on the phylum taxonomic assignment, and black squares indicate the absence of the referred metabolite/enzyme. Heatmap is clustered according to the phylogeny of the recovered MAGs. To perform the metabolic reconstruction of each MAG, the annotation obtained from KOFAM database were filtered to keep only the top 5 hits of each protein (e-value < 1e−5). These filtered annotations were supplied to the Annotation of Metabolite Origins (AMON) tool, which predicts the metabolites that each MAG can produce, according to KEGG Orthologous (KO) assignments.

abundance of this metabolite in both cecal and rectal samples (Supplementary Data 6). On the other hand, the expression analysis of key genes involved in the butyrate pathway (*atoA/D* genes) indicates that Firmicutes *Ileibacterium* sp. MAG6 and *Megasphaera* sp. MAG33 are likely the major butyrate-producing bacteria in the capybara gut (Supplementary Fig. 6 and Supplementary Data 7). The Bacteroidetes MAG47 and Fuso-bacteria MAG38 and MAG39 also have co-localized genes *atoA/atoD* and *ptb/butK*, suggesting that they might contribute to butyrate production, in some extent (Supplementary Fig. 6 and Supplementary Data 7). The typical genes from acrylate and propanediol pathways involved in propionate production were not identified in the recovered MAGs from capybara gut (Fig. 5), but the *mmdA* gene encoding a methylmalonyl-CoA decarbox-ylase from the succinate pathway, is widespread mainly among Bacteroidetes and was also observed in some Firmicutes and Fusobacteria MAGs (Supplementary Fig. 6 and Supplementary Data 7). Furthermore, the ratio of propionate detected in the capybara gut correlates ($R = 0.77$ and $p = 0.07$) with the relative

abundance of Bacteroidetes, supporting the succinate pathway from this phylum as the major source of propionate production in this environment.

The detection of high SCFAs levels in the capybara digestive tract, which is a common marker of digestion performance of dietary fibers[36] along with the identification of the several metabolic pathways involved in SCFAs production corroborates the potential of this microbiota for the breakdown of recalcitrant plant polysaccharides with concomitant production of energetic metabolites for the host.

**A β-galactosidase GH family discovered from capybara gut microbiome.** Drawing on the results showed herein, the capybara gut microbiome can be an important source of microbes har-boring uncharted enzymes involved in plant polysaccharides depolymerization. Moreover, the joint MG and MT analysis of the capybara gut microbiome revealed several expressed genes annotated as hypothetical proteins. Some of these genes display

**Table 1 Kinetic parameters of recombinant CAZymes.**

| Protein name | CAZy family | Substrate | pH | T (°C) | $K_m$ | $k_{cat}$ (s$^{-1}$) | $k_{cat}/K_m$ |
|---|---|---|---|---|---|---|---|
| CapGH97 | GH97 | pNP-α-D-Gal[a] | 7.0 | 35 | 7.70 ± 0.61 (mM) | 16.58 ± 0.49 | 2.15 |
| CapGH10 (full-length) | GH10 | Rye arabinoxylan | 5.0 | 50 | 2.09 ± 0.31 (mg mL$^{-1}$) | 162.74 ± 11.34 | 77.87 |
| | | Beechwood xylan | 5.5 | 55 | 1.61 ± 0.10 (mg mL$^{-1}$) | 16.96 ± 0.46 | 10.53 |
| CapGH10$_T$ (truncated GH10 catalytic domain) | GH10 | Rye arabinoxylan | 5.5 | 55 | 1.76 ± 0.13 (mg mL$^{-1}$) | 206.76 ± 6.85 | 117.48 |
| | | Beechwood xylan | 5.5 | 55 | 1.75 ± 0.18 (mg mL$^{-1}$) | 79.48 ± 3.16 | 45.42 |
| CapGH43_12 | GH43_12 | pNP-α-L-AraF | 6.5 | 35 | 2.74 ± 0.29 (mM) | 151.19 ± 6.21 | 55.18 |
| CapGH173 | GH173 | pNP-β-D-Gal | 7.5 | 45 | 0.57 ± 0.05 (mM) | 17.6 ± 0.39 | 30.88 |
| BXY_26070 (CBK67650) | GH173 | pNP-β-D-Gal | 7.5 | 45 | 1.19 ± 0.35 (mM) | 29.85 ± 1.95 | 25.08 |

The pH, temperature, and substrate saturation curves are shown in Supplementary Figs. 7, 12–14. The kinetic parameters, $k_{cat}$ (turnover number) and $K_m$ (Michaelis constant) are expressed as mean ± SD from three independent experiments ($n = 3$).
[a]Kinetic parameters determined using 5 mM of CaCl$_2$. Source data are provided as a source data file.

remote similarity to CAZy members, with sequence identity ranging from 10 to 21%, suggesting a potential function in the processing of plant polysaccharides, but requiring further functional investigation (Supplementary Table 1).

One of these hypothetical proteins (SEQ ID PBMDCECB_44807, named here as CapGH173), was recovered from *Bacteroidales bacterium* MAG42, a discovered genome that expands the uncultured UBA932 family (Supplementary Data 1). Biochemical characterization of CapGH173 showed it is active on *p*-nitrophenyl-β-D-galactopyranoside (pNP-β-D-Gal) and kinetic parameters were determined from substrate saturation curves (Table 1 and Supplementary Fig. 7). CapGH173 orthologues are found in Actinobacteria, Firmicutes, Verrucomicrobia and Bacteroidetes MAGs recovered from diverse sources such as rumen, feces, gut, and oral microbiotas (Supplementary Table 2), being the closest sequence from a rumen-derived MAG (UBA2817) from the uncultured *RC9* group[22]. Phylogenetic analysis showed that CapGH173 is remotely related to GH-A CAZy families, with GH30 and GH5 being the closest ones (Fig. 6a). Protein modeling and threading performed using RoseTTAFold[37] and PDBsum[38], respectively, revealed that CapGH173 consists of a (α/β)$_8$-barrel structure (Supplementary Fig. 8), which is an archetypal scaffold of the clan GH-A. According to structural predictions, CapGH173 exhibits a two-domain architecture including an appended β-sandwich domain (Supplementary Fig. 9), which is a similar structural organization found in the GH30 family. With the exception of the residues defining the clan GH-A, sequence alignment with GH5 and GH30 members revealed a very low sequence conservation below the criterium for significant similarity detection (using an *e*-value < 0.05), demonstrating that although the domains in the tertiary structure can be similar, the sequences between these families are remarkably diverse (Supplementary Figs. 9–11 and Supplementary Table 3). To further explore the discovered GH173 family, the enzyme BXY_26070 (SEQ_ID CBK67650.1) from *B. xylanisolvens*, which shares 46% sequence identity with CapGH173, was also heterologously expressed, purified, and biochemically characterized (Table 1). The two members characterized from the GH173 family present β-galactosidase activity, which is not described in either GH30 or GH5 families, strengthening at the biochemical level the establishment of this GH family.

In the *Bacteroidales bacterium* MAG42, CapGH173 is found in a predicted PUL that includes enzymes from families GH2 and GH78 (Fig. 6b). A similar PUL organization was also predicted in *Bacteroidetes* sp. 1_1_30 recovered from the human gut, which yet harbors enzymes from GH36, CE7, and PL8_2 families (Fig. 6b). It is worth to mention that CapGH173 is often found fused to a GH36 module or in PULs having GH36 members, as in *B. xylanisolvens* and *Prevotella dentalis*, recovered from the stool and oral cavity, respectively (Fig. 6b), indicating a synergistic

relationship between these families. Moreover, these families are also commonly found along with GH78 α-L-rhamnosidases in the PUL context. In *Bacteroidales bacterium* UBA2817, a GH173 member is appended to a GH78 module carrying a CBM67, both targeting rhamnogalacturonans (Fig. 6b). These observations suggest that GH173 could act on β-linked galactosyl residues in pectic polysaccharides.

**Capybara *Bacteroidaceae bacterium* MAG57 harbors an unprecedented family of carbohydrate-binding module.** As presented in the taxonomic analysis, *Bacteroidaceae bacterium* MAG57 encompasses a remarkable number of CAZyme-encoding genes including a gene cluster targeting arabinoxylan (CC102), an abundant hemicellulose in secondary cell walls of sugarcane and other grasses. This cluster encodes two exo-enzymes from families GH43 and GH97, and an unconventional GH10 member with an unknown 45 kDa N-terminal domain (Fig. 7a). Sequence analysis showed that this unusual N-terminal domain is also present in Bacteroidetes MAGs derived from the gut of human, mouse, and elephant (Supplementary Table 4); however, it displays no similarity with any known ancillary domain associated with CAZymes. Therefore, to evaluate the function of this unconventional GH10 member, the full-length protein, its domains apart, and the other GH members comprising the CC102 cluster were recombinantly expressed, purified, and characterized. The GH97 member (CapGH97) is a calcium-activated α-galactosidase, whereas the GH43 member (CapGH43_12) is a highly active α-L-arabinofuranosidase (Supplementary Figs. 12, 13 and Table 1)—two key activities to remove glycosidic substitutions of heteroxylans.

The GH10 domain of CapGH10 exhibits endo-β-1,4-xylanase activity, being active on both xylan and distinct arabinoxylans (Table 1). Kinetic analysis indicates that substitutions present in rye arabinoxylan (arabinose/xylose ratio = 40/60) are not detrimental to the catalytic performance, showing similar $K_m$ (Michaelis constant) and $k_{cat}$ (turnover number) constants compared to xylan (Table 1 and Supplementary Fig. 14). The endo-β-1,4-xylanase enzyme from *Hungateiclostridium themocellum* ATCC 27405 (Xyn10Z) is the closest characterized GH10 member, sharing 35% of sequence identity with CapGH10[39]. The N-terminal region of Xyn10Z encompasses a feruloyl esterase followed by a CBM6 domain, which is not conserved in CapGH10[40]. The CapGH10 N-terminus showed only sequence similarity with uncharacterized proteins and the closest homologs, featuring similar domain architecture, were identified in genomes from ruminal *Prevotella* sp. such as *Prevotella* sp. BP1-148, *Prevotella* sp. BP1-145, *Prevotellaceae bacterium* HUN156 and *Prevotellaceae bacterium* MN60, also likely targeting xylan-related polysaccharides (Supplementary Table 4).

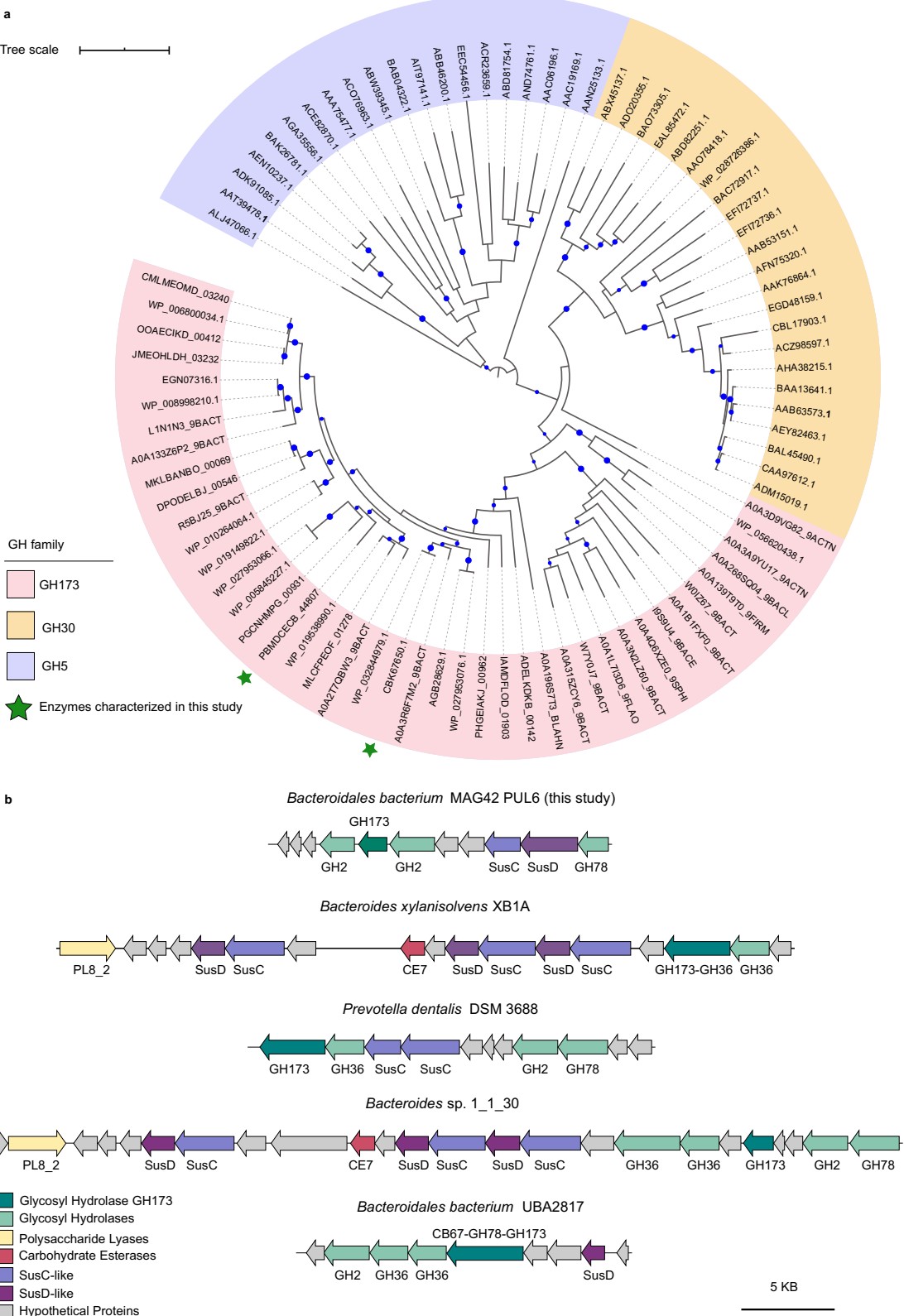

**Fig. 6 Phylogeny of the GH173 family and its genomic context. a** Maximum likelihood phylogenetic analysis of the glycosyl hydrolases GH173 family (pink background) including characterized members from the families GH30 (orange background) and GH5 (purple background). Nodes with bootstrap support values >50 are indicated by the blue circles. Founding members of GH173 family characterized in this study are denoted with a green star CapGH173 (PBMDCECB_44807) and BXY_26070 (CBK67650.1). **b** Genomic context of GH173-containing polysaccharide utilization loci (PULs) identified in Bacteroidetes metagenome-assembled genomes (MAGs).

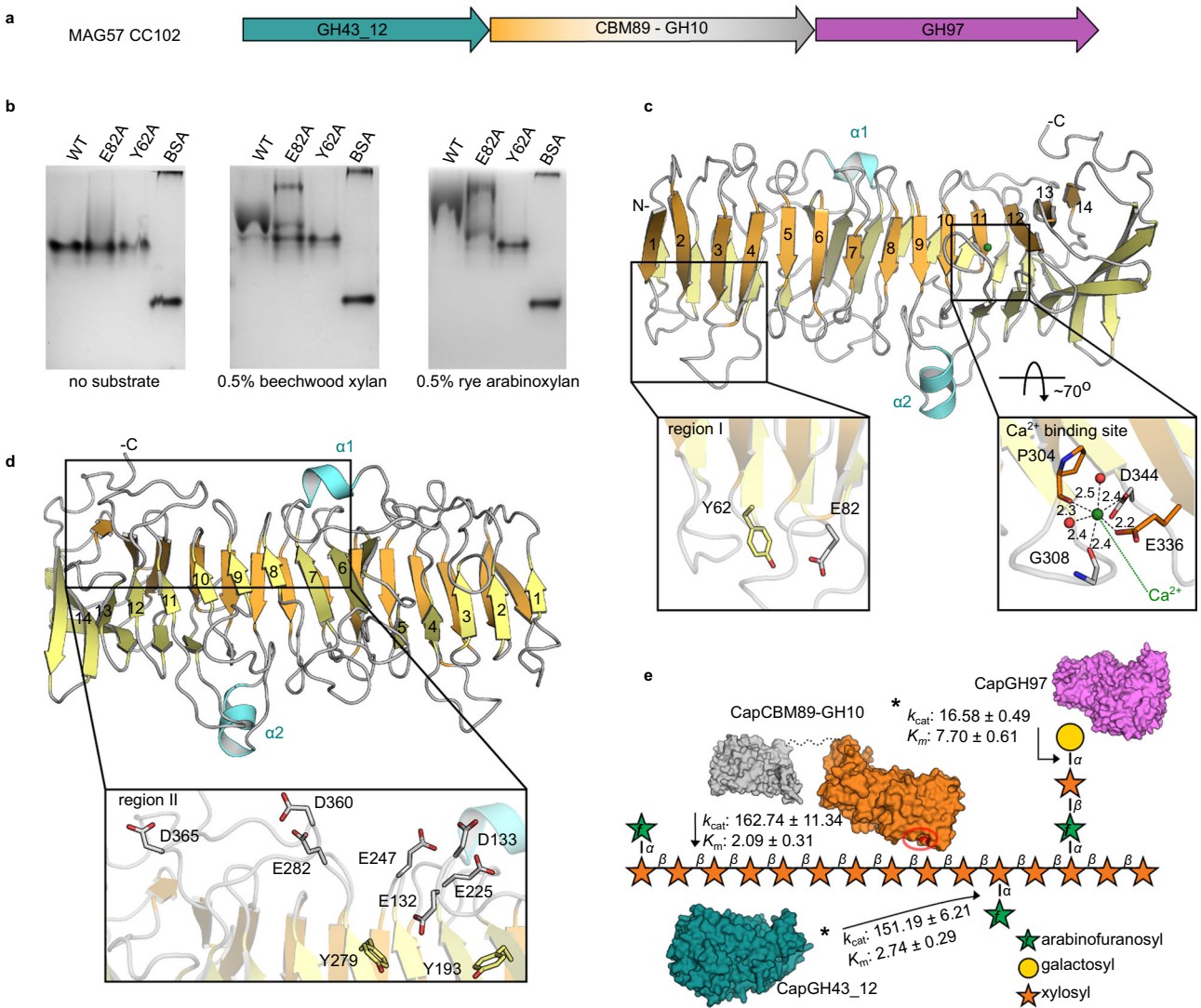

**Fig. 7 Enzymatic system for heteroxylan degradation from *Bacteroidaceae bacterium* MAG57. a** Schematic representation of the CAZyme cluster (CC102) involved in heteroxylan breakdown. **b** Affinity gel electrophoresis (AGE) of CapCBM89 (wild-type (WT) and mutants from region I) with xylan and arabinoxylan. Bovine serum albumin (BSA) was used as control. AGE experiments were independently performed three times for CapCBM89 WT ($n = 3$) and twice for mutants ($n = 2$) using arabinoxylan as substrate, with similar results. **c, d** Crystal structure of CapCBM89 highlighting the β-helix fold consisting of 14 helical turns, including the mutated residues in the regions I (**c**), II (**d**) and $Ca^{2+}$ binding site (**c**). **e** Schematic representation of the modes of action of the enzymes CapGH10, CapGH97, and CapGH43_12 on heteroxylans. The polysaccharide structure was based on information from Biely et al.[87]. $k_{cat}$ (turnover number) values are expressed in $s^{-1}$ and $K_m$ (Michaelis constant) in mM (for CapGH97 and CapGH43_12) and mg mL$^{-1}$ for CapGH10. Protein surface representations were based on molecular modeling of CapGH43_12 (blue), CapGH97 (purple), and the catalytic domain of CapGH10 (gray) using RoseTTAFold[37]. Protein surface of CapCBM89 was represented based on its crystal structure. The residues Tyr62 and Glu82, involved in (hetero-)xylan interaction, are highlighted (red) in the CapCBM89 surface (orange) and the approximated possible binding region (region I) is indicated (red circle). *Kinetic parameters of the enzymes CapGH97 and CapGH43_12 were determined using the synthetic substrates pNP-α-D-galactopyranoside (with addition of 5 mM CaCl$_2$ in the reaction) and pNP-α-L-arabinofuranoside, respectively. Source data are provided as a source data file.

The potential enzymatic activity of the isolated N-terminal domain of CapGH10 was assessed against 30 different substrates including synthetic substrates, oligosaccharides, and polysaccharides (Supplementary Table 5), but no (hydrolase, lyase, or esterase) activity was observed. Typical activities involved in heteroxylans breakdown including endo-β-1,4-xylanase, β-xylosidase, α-L-arabinofuranosidase, α-D-galactosidase, α-D-glucuronidase, 4-O-methyl-glucuronoyl methylesterase, feruloyl esterase, and acetyl xylan esterase were assayed by distinct methods without the detection of product formation or substrate consumption. Under this perspective, we then interrogated the capacity of this N-terminal domain to bind potential substrates of its GH10 partner such as beechwood

xylan and arabinoxylans using affinity gel electrophoresis (AGE). As shown in Fig. 7b, this domain can indeed interact with the substrates of the GH10 domain, suggesting that this N-terminal domain may target the CapGH10 catalytic domain to xylan polysaccharides.

To get further insights into the potential role of this unconventional N-terminal domain, its crystallographic structure was solved by SeMet phasing at 1.8 Å resolution (Supplementary Table 6). The domain behaves as a monomer in solution (Supplementary Fig. 15) and exhibits a parallel right-handed β-helix fold, consisting of 14 complete helical turns with two main short helices protruding from the β-helix backbone (Fig. 7c). The

14 helical turns are twisted and curved with a calcium ion between the 11th and 12th turns in an octahedral coordination sphere (Fig. 7c). This β-helix fold is observed in the clan GH-N of the GH superfamily, in the carbohydrate esterase CE8, and in several polysaccharide lyase (PL) families; however, structural comparisons with these CAZy families (GH28, GH91, PL6, and CE8) led to high rmsd values (>2.9 Å), indicating poor three-dimensional conservation (Supplementary Table 7). Neither the catalytically relevant residues nor the active site topology of these families is conserved in the CapGH10 β-helix domain (Supplementary Fig. 16), supporting that this domain is not catalytically active.

In order to elucidate the molecular determinants for xylan binding observed in AGE experiments, two surface regions populated with aromatic and acidic residues, typical platforms for carbohydrate interaction, were identified and mutated. The region I between the turns 1–4 and the region II near to turns 6–10 (Fig. 7c, d and Supplementary Fig. 17). Mutations E247A and E282A, in the region II, severely impaired protein stability and led to the expression only as inclusion bodies. Mutation D344L (at the calcium-binding site) also affected protein stability, but the arabinoxylan/xylan binding capacity was preserved (Supplementary Fig. 18). This result indicates that calcium ion has a structural relevance rather than a functional role in carbohydrate recognition. Only the mutants Y62A and E82A, affected the migration pattern in AGE assays with beechwood xylan and rye arabinoxylan (Fig. 7b). Both residues are located at the region I, indicating that this patch plays a role in carbohydrate binding. It is worth to mention that two aromatic residues considered critical for the activity of a GH28 member are present in the corresponding region of the CapGH10 β-helix domain, Y193 and Y279; however, their alanine mutation did not alter the carbohydrate binding, in agreement with the lack of catalytic function of the CapGH10 β-helix domain (Supplementary Figs. 16 and 18). Combining the biochemical, structural, and mutagenesis analyses, we would define CapGH10 β-helix domain as a CBM, therefore, establishing a distinguishing structural scaffold in this superfamily and founding the family CBM89.

The combination of this unconventional modular endo-β-1,4-xylanase with complementary GH43 and GH97 enzymes constituting the cluster CC107 likely confer the ability to *Bacteroidaceae bacterium* MAG57 to act on complex heteroxylans (Fig. 7e), a key function in the gut microbiome of capybara that have grasses as a major component in its diet.

## Discussion

In this study, we investigated how the gut microbiota of the largest living rodent, capybara (*Hydrochoerus hydrochaeris*) known as "master of the grasses", can efficiently depolymerize and utilize recalcitrant plant polysaccharides. These semi-aquatic animals are hindgut fermenters throughout found in Pantanal wetlands and Amazon basin and, in particular, such animals dwelling the Piracicaba basin region in Brazil have incorporated sugarcane in their diet for decades, a relevant timescale for microbial adaptation and specialization[41], reasoning that this microbiota has been shaped and optimized for energy extraction[21] from this industrially relevant lignocellulosic bio-mass. Sugarcane is an important feedstock for Brazilian economy and other countries such as India and Thailand. Two-thirds of this crop are made of lignocellulose, which currently is left in the field (straw) and burnt (bagasse) for energy purposes (electricity and vapor). The understanding of the enzymatic and metabolic mechanisms employed by these microbial communities to obtain energy from plant fibers may unveil alternative biological systems for the conversion of these lignocellulosic agro-industrial residues into value-added products and create further opportunities for carbohydrate-based biotechnological applications.

An integrated meta-omics approach focused on the enzymatic and metabolic capabilities for plant fiber breakdown unveiled that cellulose degradation in this community is not accomplished by classical mechanisms involving cellobiohydrolases or cellulosomes. Instead, cellulose is likely processed by the sophisticated mechanisms from Fibrobacteres, including single and multi-modular CAZymes secreted by the T9SS system, CAZyme-rich outer membrane vesicles, and lignocellulose adhesion proteins (Fig. 8). The complex and diverse composition of hemicellulosic and pectic polysaccharides present in gramineous and aquatic plants are tackled by a broad number of CAZymes organized in PULs found in the multiple recovered Bacteroidetes MAGs, which in part resembles to that from human gut Bacteroidetes species such as the PULs for mixed-linkage β-glucans[29] and xyloglucans[42,43]. From one of these Bacteroidetes MAGs, which represents a taxonomic novelty (MAG57), an elaborate CAZyme cluster targeting heteroxylans was biochemically elucidated, which may contribute to circumvent the recalcitrance of these polysaccharides.

The CAZyme repertoire of the capybara gut microbiome comprehensively covers the most abundant and recalcitrant polysaccharides present in gramineous and aquatic plants, which also requires efficient metabolic capabilities to further convert these depolymerized polysaccharides into SCFAs, the main energy source of the host. This hypothesis was validated by metabolite profiling and metabolic reconstructions with the identification of acetate, butyrate, and propionate as the main metabolites, produced by classical sugar-to-SCFAs metabolic pathways including pyruvate-Acetyl-CoA for acetate, succinate for propionate, and both butyryl-CoA:acetate-CoA-transferase and phosphotransbutyrylase/butyrate kinase for butyrate (Fig. 8). Similar microbial strategies for SFCAs production were also observed in human gut bacteria, highlighting additional commonalities between the gut microbiota from omnivores and hindgut fermenters[44].

It was prominent the identification of genes and PULs with remote or no similarity to known CAZy families and systems, which led to the discovery of two CAZy families including a high-molecular-weight CBM family involved in (hetero-)xylan recognition (CBM89) and a GH-A clan family of β-galactosidases (GH173). CBM89 is an unconventional CBM family featuring a β-helix fold, demonstrating that such domains, through the evolution, were also sculpted from tandem repeat structures, which could be exploited as a versatile platform for the rational design and engineering of CBMs. The GH173 family expands the panel of industrially relevant enzymes since β-galactosidases are broadly employed in the food and beverage industries, especially in the processing of dairy-based products.

In conclusion, this work sheds light on the enzymatic apparatus and metabolic pathways employed by the gut microbiota from the Amazon monogastric semi-aquatic herbivore, capybara, for the breakdown and utilization of recalcitrant dietary polysaccharides. The discovery of several taxonomic novelties associated with plant fiber degradation along with the founding of two CAZy families highlight this microbiota as an untapped source of CAZymes and enzymatic systems of biotechnological interest. Furthermore, this comprehensively and multidisciplinary investigation of the capybara microbiome advances our understanding regarding the molecular strategies exploited by the microbiota of hindgut herbivores to utilize complex plant glycans as an energy source, indicating that the exploration of such ecological niches might create opportunities to leverage sustainable carbohydrate-based technologies.

## Methods

**Procedures for sample collection**. This study was carried out in strict accordance with the Animal Management Rule of the Brazilian Ministry of Environment

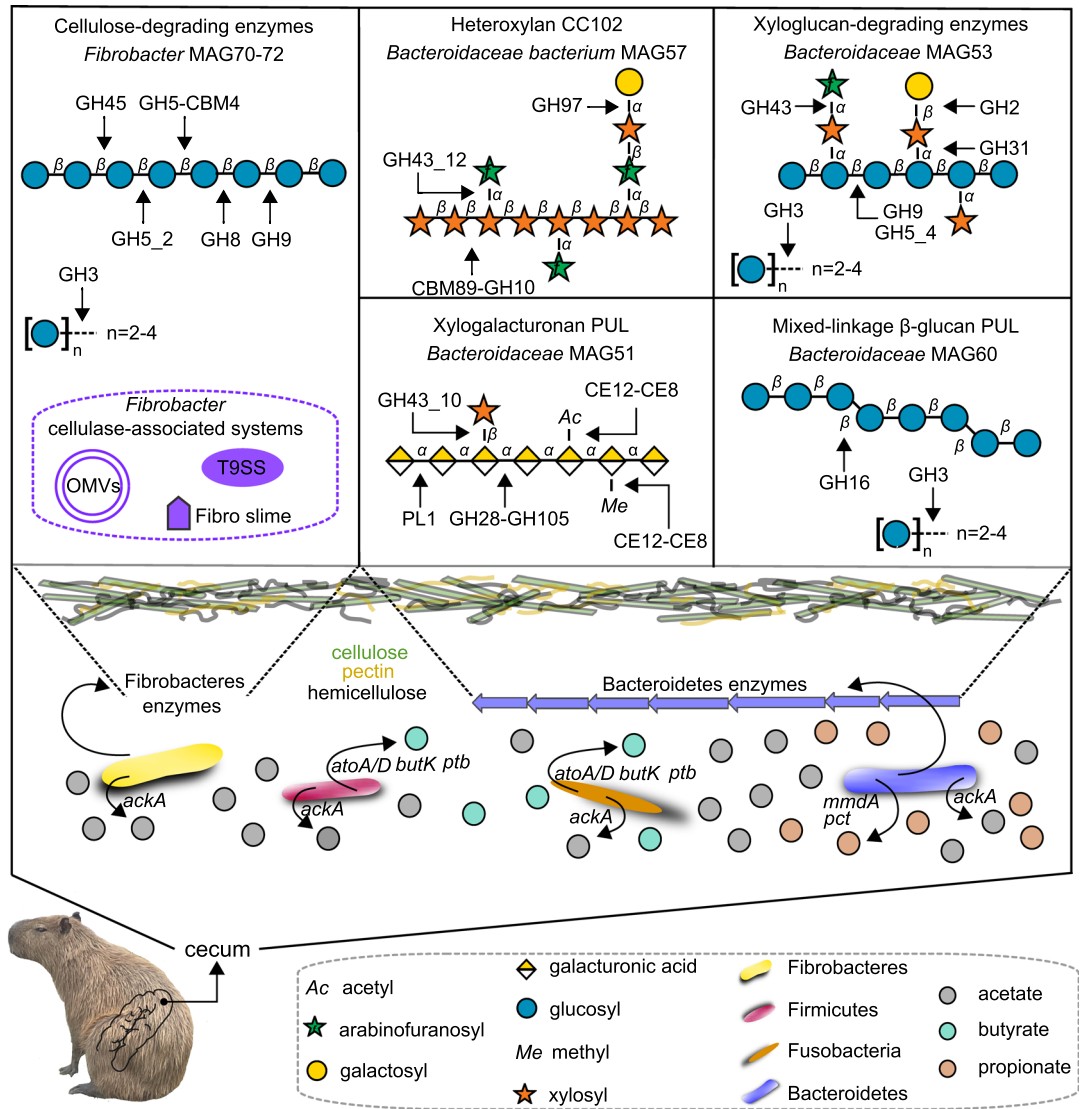

**Fig. 8 Schematic representation of the capybara gut microbial community and enzymatic strategies involved in the depolymerization and conversion of dietary polysaccharides into short-chain fatty acids (SCFAs).** In the upper panel is highlighted the CAZymes and mechanisms accounted to the depolymerization of cellulose, hemicellulose, and pectins usually found in grasses. Simplified schematic representations of cellulose, heteroxylan, xyloglucan, xylogalacturonan, and β-glucan based on information from Zugenmaier[88], Biely et al.[87], Pauly and Keegstra[89], Harholt et al.[90] and Izydorczyk and Dexter[91], respectively. In the lower panel is shown the key phyla associated with hexoses and pentoses conversion into SCFAs. Genes encoding products *ackA*: acetate kinase; *butK*: butyrate kinase; *ptb*: phosphate butyryltransferase; *atoA*: Butyryl-CoA:acetate-CoA-transferase; *atoD*: Butyryl-CoA:acetate-CoA-transferase; *pct*: propionate CoA-transferase; *mmdA*: methylmalonyl-coa decarboxylase.

(Sisbio 59826-1). The samples were collected from three euthanized young female animals in Tatuí/São Paulo State, Brazil (September 2017), which were not infected with *Rickettsia rickettsii* assessed by immunofluorescence assays[45], as a control procedure of Rocky Mountain Spotted Fever (RMSF) hosts. After euthanasia, the animals were submitted to abdominal surgery to collect fresh samples (20 g) from the cecum and recto of each animal. All samples were placed in sterile containers and immediately frozen in liquid nitrogen. Samples were kept at −80 °C until processing.

**Microbial DNA and RNA extraction.** Samples of cecal and rectal contents were frozen in liquid nitrogen and pulverized with an oscillating ball mill (TE-350, Tecnal Inc.). The homogenized samples were used for microbial DNA extraction according to the protocol described by Yu and Morrison[46] with modifications. Briefly, 0.25 g of sample was transferred to a Lysing Matrix E Tube from the FastDNA Spin Kit (MP Biomedical, Inc.). For cell lysis, 1 mL RBB+C buffer was added in each sample, followed by homogenization in a FastPrep® FP120 instrument (MP Biomedical, Inc.). The precipitation of proteins was carried out with the addition of 0.26 mL of 10 M ammonium acetate followed by incubation on ice for 5 min and centrifugation at 4 °C for 10 min at 16,000 × *g*. The precipitation of nucleic acids was performed with isopropanol (1 mL). The samples were incubated on ice for 30 min and then centrifuged at 4 °C for 15 min at 16,000 × *g*. The

pellet was recovered and washed with 70% (v/v) ethanol, followed by drying at room temperature. The nucleic acid pellet was dissolved in 75 μL of autoclaved ultrapure water. RNA was removed with the addition of DNase-free RNase (2 μL from a 10 mg mL⁻¹ stock solution). DNA purification was performed using PowerClean® DNA Clean-Up Kit (Mo Bio Laboratories). Finally, electrophoresis using 0.8% (w/v) agarose gel was used to separate the DNA fragments and to evaluate DNA quality. The DNA solution was stored at –20 °C.

The homogenized samples with the oscillating ball mill were also used for RNA extraction. In this experiment, 500 mg of each sample was used for total RNA extraction with Trizol and FastRNA® Pro Green Kit (MP Biomedicals). The quality of RNA was verified using an Agilent Bioanalyzer 2100 with the RNA 6000 Nano Reagents Kit (Agilent) and RNA samples with RNA integrity number (RIN) > 8.0 were treated with a blend of the Ribo-Zero rRNA removal Kit for bacteria and the Ribo-Zero Magnetic Gold Kit (Epicentre Biotechnologies) to remove both prokaryotic and eukaryotic rRNAs, respectively. Subsequently, the supernatant was purified using an 80% (v/v) ethanol solution, and the resultant RNA were used for library RNA preparation.

**Microbial community structure and diversity analysis.** Capybara gut microbial community structure and diversity was investigated via high-throughput sequencing of 16S rRNA gene. The amplification of the 16S rRNA gene V4 region was performed in

technical and biological triplicates using the 515F (5'-GTGCCAG CMGCCGCGGTAA) and 806R (GGACTACHVGGGTWTCTAAT) primers[47]. Sequencing was performed on a MiSeq Sequencing System (Illumina Inc.) with the V3 kit, 600 Cycles, in paired-end sequencing mode $2 \times 300$ bp. The ZymoBIOMICS™ Microbial Community DNA Standard (D6305, Zymo Research) with eight phylogenetically distant bacterial strains (3 gram-negative and 5 gram-positive) and 2 yeasts, was included as a positive control to evaluate possible bias in libraries construction, sequencing, and bioinformatics analysis. For taxonomic analysis, paired-end reads were quality checked using FastQC (https://www.bioinformatics.babraham.ac.uk/projects/fastqc/) and filtered using Trimmomatic v.0.36[48] to remove adapters and low-quality reads, using the following parameters: ILLUMINACLIP:NexteraPE-PE.fa:2:30:10:8:true, LEADING:4, TRAILING:4, MINLEN:60 and SLIDINGWINDOW:4:16. Filtered paired-end reads were merged using fastq_mergepairs function from Usearch v.10 package[49] (parameters: fastq_maxee 0.5, fastq_minovlen 50, and fastq_minmergelen 250). Detailed number of reads per sample are summarized in Supplementary Table 8. Furthermore, primer sequences were removed, and singletons were discarded. Filtered amplicon reads were denoised (error-corrected) using the UPARSE unoise3 function (parameters: -minsize 8 and alpha 0.2), to likely recover true biological sequences (zOTUs – zero radius OTUs). Prokaryotic taxonomic assignment was performed using sintax function, as implemented in Usearsh v.10, using a the sintax_cutoff parameter of 0.8 as threshold and the RDP database v16[50]. Further analyses were performed using phyloseq v1.20 package on the R Studio. Pearson and Kendall correlations were calculated between taxonomic assignments performed with 16S, MG, MT, and 16S from MG. A significant Pearson correlation ($P < 0.05$) was observed for all omics (correlation coefficient $r = 0.96$ for MG:MT, $r = 0.74$ for 16S_MG:MG and $r = 0.71$ for 16S_MG:MT, $P < 0.05$), except when comparing with 16S data.

**Metagenome and metatranscriptome sequencing**. Metagenomic libraries were prepared using the Nextera Library Preparation kit (Illumina Inc.), while metatranscriptomic libraries were prepared using the TruSeq Stranded total RNA Library Prep Kit (Illumina Inc.). Libraries concentrations were measured through quantitative qPCR using the KAPA Library Quantification Kit (Roche Inc.) and assayed for quality using an Agilent Bioanalyzer 2100 system (Agilent Technologies). MG and MT libraries were paired-end sequenced in two runs ($2 \times 100$ bp) on the Illumina HiSeq 2500 platform at the NGS sequencing facility at LNBR/CNPEM (Campinas, Brazil). Furthermore, the cecal and rectal gDNA were homogenized into a single sample and sequenced on a MinION sequencing device (Oxford Nanopore Technologies Inc.) to obtain long reads. About 1 µg of ultra-long high-molecular weight gDNA from the homogenized samples was used for library preparation using the SQK-LSK109 Kit (Oxford Nanopore Technologies Inc.). The MinION run was performed on a Flow cell R9 version, generating around 3 Gb of long reads.

**Metagenome and metatranscriptome analysis**. MG and MT raw sequences were quality-checked and trimmed as described above. MT reads were also analyzed using SortmeRNA v. 2.0 to remove rRNA reads, and then both MG and MT reads were taxonomically classified using Kaiju v.1.7.4 with a maximum number of mismatches allowed = 5 and with the greedy mode[51]. Reago was used to recover 16S ribosomal RNA from the MG data[52]. For functional analysis, the MG trimmed reads were de novo co-assembled using IDBA_UD (version 1.1.1) with the pre-correction parameter and k-mer size from 20 to 60[53]. Assembly statistics are described in Supplementary Table 9. The assembled MT was binned using CONCOCT v.0.4.0 (parameters: -c 400, -k 4 -l 1000, -r 200 and --no_cov_normalization)[54] and MaxBin 2.0 (parameters: min_contig_length 1000, max_iteration 50, prob_threshold 0.9 and markerset 107)[55] to recover putative genomes from the MT data. The binned genomes were dereplicated to remove redundancies using dRep v. 2.0.5 (parameters: -comp 80 -con 10 -str 100 and -p 10) and analyzed using CheckM v1.0.6 with the lineage_wf workflow[56] to determine the completeness and contamination ratios of these genomes. Long-reads sequencing (ONT) were used for MAGs scaffolding using SSPACE-long-reads v1.1 (parameters: -k 5, -a 0.7, -x 1, -m 50, -o 20 and -n 1000), resulting in 24 MAGs of high quality (completeness > 90% and contamination < 5%) and 50 MAGs of medium quality (completeness > 50% and contamination <10%), according to parameters proposed by Bowers et al. 2017[57] (Supplementary Data 1). Genomes with completeness lower than 55% and more than 15% contamination rate were discarded. To assign taxonomy to the recovered genomes, GTDB-tk tool v.1.4 was used with the release 202 of the GTDB database[58]. Gene prediction and annotation of both the recovered genomes and the co-assembly were performed using Prokka v.1.11 with the meta parameter[59], and annotation statistics are described in Supplementary Table 8. KEGG Orthologous (KOs) and pathways annotation were performed using KOFAM (e-value < 1e−5)[60] and functional ontology assignments for metagenomes (FOAM) database (e-value < 1e−5)[61]. CAZymes and PULs annotations were performed according to pipelines established and developed by the CAZy database team, based on HMM search profiles and sequence similarity to known CAZymes[62,63]. Furthermore, EC number and KO annotations were also inspected and provided (Supplementary Data 4) to assure proper annotation of CAZymes[62]. Furthermore, MG and MT reads were mapped to the whole set of genes recovered from the co-assembled MT and the set of genes recovered from the MAGs using Kallisto v. 0.46.1 with quant function[64] to estimate the coverage/

abundance of protein-coding genes in cecal and rectal samples. Normalized abundance was estimated based on the count/number of reads per kilobase per million mapped reads and expressed as TPM (Transcripts Per Million).

**Phylogenetic analysis and metabolic reconstruction**. Phylogenetic analysis of the MAG57, reference *Bacteroidetes* type strains, and *Prevotellaceae* uncultured genomes recovered from UBA project[22] was performed using concatenated 92 single-copy core genes according to UBCG method[65]. CAZymes phylogenetic analysis was carried out using the catalytic domain of each family aligned with MAFFT[66], and using maximum likelihood methods implemented in the RAxML software[67], with 1000 rapid bootstrap inferences and LG as the substitution model.

To perform the reconstruction of metabolic pathways of each recovered MAG, their annotation obtained from the KOFAM database were filtered to keep only the top 5 hits of each protein with e-value below the 1e−5 threshold. These filtered annotations were then supplied to the Annotation of Metabolite Origins (AMON) tool[68], which based on the KOs annotated in each MAG predicts the putative metabolites that it can generate.

**NMR-based metabolomics**. Approximately 30 mg of dried cecal and rectal contents, and 300 µL of solution 2:1 (methanol: chloroform) were mixed and sonicated for 1 min (4 cycles of 15 s with intervals of 10 s) and placed at 4 °C for 15 min. Next, 300 µL of solution 1:1 (methanol: ultrapure water) was added, followed by centrifugation at $16,000 \times g$ and 4 °C for 20 min. The supernatant was transferred to another tube and were dried in a CentriVap Solvent System (Labconco Corporation). Samples were diluted to 630 µL by addition of $D_2O$, 70 µL of sodium phosphate buffer (final concentration 0.1 M) containing dimethyl-silapentane-sulfonate (final concentration 0.5 mM) for NMR chemical shift reference and concentration calibration. The samples were filtrated in a syringe filter with a 0.22 µm pore size hydrophilic polyethersulfone (PES) membrane. The final volume of filtrate ranged from 500 to 650 µL. 1H NMR spectra of samples were acquired using a Varian Inova NMR spectrometer (Agilent Technologies Inc.) equipped with a 5 mm triple resonance cold probe and operating at a 1H resonance frequency of 599.84 MHz and constant temperature of 298 K (25 °C). A total of 1024 free induction decays were collected with 32-k data points over a spectral width of 16 ppm. A 1.5-s relaxation delay was incorporated between scans, during which a continual water presaturation radio frequency (RF) field was applied.

The metabolites were processed and quantified using the NMR Suite software version 7.5 (Chenomx Inc™, Edmonton, AB, Canada). *Processor* module of this software was used to adjust the spectral phase and baseline corrections. A 0.5 Hz line-broadening function was used to reduce signal noise and facilitate the fitting of the metabolite signals in spectral peaks. The water signal was suppressed, and the spectra were calibrated using the reference signal of the TMSP-$d_4$ as 0.5 mM. The spectra were individually transferred to the *Profiling* module of this software to determine the metabolomic profile of each group. Metabolites were identified and their concentrations were measured. Metabolite concentration data were normalized using the initial amount of extraction.

**Protein expression and purification**. Protein expression and purification were conducted as reported in Santos et al.[69]. Briefly, *E. coli* BL21 strain was transformed with target genes subcloned into the pET28a vector in frame to a 6×His-Tag at the N-terminus. Transformed strains were grown in selective LB medium (0.5% (w/v) yeast extract, 1% (w/v) tryptone, 1% (v/v) sodium chloride) at 37 °C until the O.D.$_{600nm}$ around 0.8. Then, the temperature was decreased to 18 °C and protein expression was induced with the addition of 0.2 mM isopropyl β-D-1-thiogalacto-pyranoside (IPTG) (Sigma Aldrich) and incubated for 16 h. Cells were then harvested by centrifugation at 5000×g for 15 min and further the pelleted cells were resuspended in saline-phosphate buffer (20 mM sodium phosphate, 500 mM NaCl, pH 7.5) containing 5 mM imidazole, 1 mM phenylmethylsulfonyl fluoride (PMSF), 5 mM benzamidine and 0.1 mg mL$^{-1}$ lysozyme. Cells were then disrupted by sonication (20-30 cycles of 15 s with intervals of 20 s), followed by centrifugation at $30,000 \times g$ and 4 °C for 60 min. The soluble protein lysates were applied to a 5-ml HiTrap Chelating HP column (GE Healthcare) and 6×his-tagged target proteins were eluted an imidazole gradient up to 0.5 M. 6×His-Tag was cleaved using 1% (w/w) trypsin (catalog no. T1426, Sigma Aldrich). Target proteins were further purified by size-exclusion chromatography with a HiLoad 16/600 Superdex 75 pg column (GE Healthcare) equilibrated with 20 mM sodium phosphate, 150 mM NaCl, pH 7.5. Purified proteins were evaluated by denaturing polyacrylamide gel electrophoresis (DLS) and samples with high purity (>95%) and low polydispersity (<20%) were employed in biochemical and biophysical experiments.

**Enzyme assays**. Purified enzymes were screened against polysaccharides, oligo-saccharides, and synthetic substrates as described in Supplementary Table 2. The saturation curves of the enzymes CapGH43_12 (α-L-arabinofuranosidase, 35 °C and pH 6.5), CapGH97 (α-galactosidase, 35 °C and pH 7.0) and CapGH173 (β-galactosidase, 45 °C and pH 7.5) were obtained using the synthetic substrates $p$-nitrophenyl-α-L-arabinofuranoside (pNP-α-L-AraF), $p$-nitrophenyl-α-D-galactopyranoside (pNP-α-D-Gal) or $p$-nitrophenyl-β-D-galactopyranoside (pNP-β-D-Gal) (Sigma Aldrich), respectively, at the optimal pH and temperature of each enzyme in McIlvaine buffer

(70 mM). The saturation curves of the full-length CapGH10 enzyme (50 °C and pH 5.0 or 5.5) and of the isolated GH10 catalytic domain (55 °C and pH 5.5) were determined for both beechwood xylan and rye arabinoxylan substrates using the 3,5-dinitrosalicylic acid method[70] in McIlvaine buffer (70 mM). Kinetic data were calculated from initial velocities and expressed as mean ± SD from three independent experiments ($n = 3$) using OriginPro v 8. Binding capacity of the CBM89 domain from the CapGH10 enzyme and its mutants were evaluated by affinity gel electrophoresis (AGE) according to Mandelli et al.[71]. Briefly, continuous native polyacrylamide gels consisted of 7.5% (w/ v) acrylamide in 25 mm Tris, 250 mM glycine buffer (pH 8.3) with 0.5% (w/w) of each polysaccharide. 10 µg of the WT enzyme/mutants and BSA (negative control) were loaded on the gels and subjected to electrophoresis at room temperature for 2 h and 60 mA. Proteins were visualized by Coomassie Blue stain.

**Small-angle X-ray scattering (SAXS)**. SAXS data of the CBM89 domain from the CapGH10 enzyme were collected at the SAXS1 beamline (Brazilian Synchrotron Light Laboratory, Campinas, Brazil) at a protein concentration of 8.4 mg mL$^{-1}$ in 20 mM Hepes buffer pH 7.5. Buffer scattering were recorded and subtracted from the protein scattering. SAXS patterns were integrated using Fit2D[72] and GNOM[73] was used to evaluate the pair-distance distribution functions $p(r)$. Ab initio molecular envelope was calculated from SAXS data with DAMMIF[74] and the crystallographic coordinates were fitted into the SAXS low-resolution model using SUPCOMB[75].

**Crystallization, X-ray diffraction, structure determination, and molecular modeling**. Crystallization experiments of the CBM89 domain from the CapGH10 enzyme were carried out by the sitting-drop vapor-diffusion method at 18 °C. Native crystals were grown in 20% (w/v) PEG6000, 0.1 M sodium acetate (pH 5.0) and 0.2 M sodium chloride. SeMet crystals were obtained under the same condition with the addition of 20 mM betaine hydrochloride. For cryoprotection, crystals were soaked in the reservoir solution added with glycerol or PEG400 (20% (w/v)) prior to flash cooling. Diffraction datasets of native and SeMet crystals were collected at the PROXIMA-2A and MX2 beamlines from SOLEIL (Gif-sur-Yvette Cedex, France) and LNLS (Brazilian Synchrotron Light Laboratory, Campinas, Brazil), respectively. Datasets were indexed, integrated, merged, and scaled using the XDS package[76]. The structure was solved by single anomalous dispersion (SAD) using the programs SHELXC, SHELXD, and SHELXE[77] for data preparation, anomalous scatters location and phase calculation, respectively. An initial model was built with the AutoBuild Wizard[78] from the Phenix package[79]. The structure was refined with the programs PHENIX.REFINE[80] and REFMAC5[81], and the models were inspected and manually adjusted according to the computed σ$_A$-weighted $(2F_0 - F_c)$ and $(F_0 - F_c)$ electron density maps using COOT[82]. TLS groups were calculated by TLSMD[83] and applied during the crystallographic refinement. The refined structure was evaluated with the servers MolProbity[84] and the PDBRedo[85]. Structure factors and atomic coordinates were deposited at the Protein Data Bank (PDB) under the accession codes 7JVI. Data collection and refinement statistics are summarized in Supplementary Table 6. Structural models of the CapGH173, GH10 domain from the CapGH10, CapGH97, and CapGH43_12 were obtained using RoseTTAFold[37] available in the Robetta structure prediction server. Protein topology of CapGH173 was obtained using PDBsum[38].

**Reporting summary**. Further information on research design is available in the Nature Research Reporting Summary linked to this article.

## Data availability

All sequencing data generated in this study can be found under the BioProject ID PRJNA563062. The 16S, metagenomic and metatranscriptomic reads for cecal and rectal samples have been deposited in the Sequence Read Archive (SRA) under the accession numbers SRR11852069-SRR11852086, SRR11852046-SRR11852057, and SRR11852097-SRR11852108, respectively (Supplementary Table 10). The recovered MAGs have been deposited in the GenBank under the accession numbers JABUSA000000000-JABUVA000000000 (Supplementary Table 11). The NMR metabolomics data have been deposited in the Metabolomics Workbench database under accession number ST001945. Atomic coordinates and structure factors have been deposited in the Protein Data Bank (PDB) under accession code 7JVI (CapCBM89). Other data generated or analyzed during this study are included in this published article and its Supplementary Information files. Source data are provided with this paper.

## Code availability

Data and code used for microbiome analyses are publicly available at https://github.com/gpersinoti/capybara_microbiome[86].

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

## Acknowledgements

We acknowledge the Brazilian Synchrotron Light Laboratory (LNLS) for the provision of time on the MX2 and SAXS1 beamlines, the Brazilian Biosciences National Laboratory (LNBio) for the use of the crystallization (Robolab), NMR and spectroscopy facilities, and the Brazilian Biorenewables National Laboratory (LNBR) for the use of the characterization of macromolecules and next-generation sequencing facilities. LNLS, LNBio, and LNBR are operated by the Brazilian Center for Research in Energy and Materials for the Brazilian Ministry for Science, Technology, and Innovations. We acknowledge SOLEIL for the provision of synchrotron radiation facilities at PROXIMA-2A (proposal 20181915) and we would like to thank William Shepard and Martin Savko for assistance in using the beamline. We acknowledge the support from Alana H. S. Alvarenga in protein purification and solubilization assays. We are also thankful to Prof. Anne S. Meyer from the Technical University of Denmark and Prof. Lucimara M. C. Cordeiro from the Federal University of Parana for providing purified polysaccharides for enzyme assays. This research was supported by grants from Fundação de Amparo à Pesquisa do Estado de São Paulo (grant no. 2015/26982-0 to M.T.M. and postdoctoral fellowship 2016/19995-0 to M.A.B.M) and Conselho Nacional de Desenvolvimento Científico e Tecnológico (CNPq) (grant no. 305013/2020-3 to M.T.M., 408600/2018-7 to G.F.P, 439195/2016-0 to L.C, 150552/2017-3, and 142332/2017-8 to M.P.M.). L.C, "Coordenação de Aperfeiçoamento de Pessoal de Nível Superior" (CAPES; Finance code 001).

## Author contributions

L.C., G.F.P., M.A.B.M. and M.T.M. designed the study and wrote the paper. G.F.P., L.F.M., N.T., V.L. and B.H. performed the multi-omics data analyses. L.C., D.A.P., M.P.M. and M.C. performed the 16S, metagenomics, and metatranscriptomics experiments. L.C., M.P.M., M.C., M.N.D., R.A.S.P., W.C.G. and C.A.S. expressed and purified the enzymes and performed the functional characterization. L.C., M.P.M., M.A.B.M., W.C.G. and M.T.M. performed the structural analysis. L.C. and M.L.S. performed the metabolomics analysis.

## Competing interests

The authors declare no competing interests.
