## [Peer Review File · Nature Communications]

REVIEWER COMMENTS

Reviewer #1 (Remarks to the Author):

In this manuscript, Cabral et al. provide a tour de force analysis of a mammalian gut microbiome, going from metagenomic analysis to X-ray crystallography. Along the way, they discover new subfamilies of carbohydrate binding modules (CBMs) and glycoside hydrolases (GHs). The manuscript is generally well written and easy to read, if not a bit wordy and in need of a little brushing up on the English. I have no major concerns with this paper, only minor comments, including:

1. The use of the word "deconstruction" (e.g., in line 34 but recurs throughout the text) is not ideal; probably better to use "degradation" or "depolymerization."
2. "Exploring the genomic dark matter..." is a wonderful turn of phrase but perhaps a little much for a scientific manuscript.
3. Some abbreviations are used without definition (e.g., MG and MT in line 96).
4. In lines 130-137, are these differences statistically significant?
5. In lines 333-335, is this shown or just inferred?
6. In line 342, I don't understand "valorization."
7. In lines 346-348, some comment on the relative timescales of microbiome shaping versus evolution would be useful here.

Reviewer #2 (Remarks to the Author):

Cabral and colleagues elucidated the microbial community composition of capybara gut, the largest living rodent that processes lignocellulosic biomass. The authors also unveiled the enzymatic systems and metabolic pathways involved in the conversion of recalcitrant dietary fibers into short-chain fatty acids using a multidisciplinary approach, including multi-meta-omics and enzymology. The work also provides the identification and structural characterization of two novel CAZy families: a glycosyl hydrolase (GH) and a carbohydrate binding module (CBM). Because of its novelty and importance, I recommend publishing this manuscript in Nature Communications.

Please, find below some comments and suggestions:

Major:

1. Please, explain in more detail what are the main differences between the new GH family from the distantly related GH5 and GH30 families using sequence alignments, secondary structure comparisons and enzymatic activities.
2. Please, expand the overall discussion further – e.g. relevance of the findings to the industrial application of sugarcane processing? This application is mentioned in the introduction and just briefly (one sentence) in the discussion.

Minor:

1. In the abstract the sentence “combination of unique enzymatic mechanism from Fibrobacteres to degrade cellulose with a broad arsenal of CAZymes organized in PULs from Bacteroidetes” is confusing. Please, re-write.
2. On page 4, line 86: define CAZy for the first time and not on page 6 line 142.
3. Figure 4 is first introduced after Figure 1 on page 4, line 109 and on page 5, line 127. The authors could have this discussion later when Figure 4 is presented.

4. On page 5, line 116: specify the extended form for GTDB.

5. The authors suggest that the cellulose degradation in the capybara gut might be mainly accomplished by endo-beta-1,4-glucohydrolases (EC 3.2.1.4) from families GH5, GH8, GH9 and GH45. However, the abundance of these GH families is quite low (Figure 2). Is this a common feature in other microbiota animals that eat grass or it is only found in capybara?

6. On page 7, line 179: remove “as reported in”.

7. Please, mention the software used to perform the prediction of the GHXXX family structural topology (Suppl. Fig. S7 and page 9, line 246).

8. Can the authors explain why the catalytic efficiency of the single GH10 domain is better than the full-length protein against rye arabinoxylan?

9. Figure 6 or the text should be reorganized. In the text, the binding experiments of CapCBMXX are described before the crystal structure and in the Figure the other way around.

10. On page 12, line 328. GH28 active site or GH10 active site?

Reviewer #3 (Remarks to the Author):

To the authors:

The original research manuscript entitled “Gut microbiome of the largest living rodent harbors unprecedented enzymatic systems to break down complex plant polysaccharides”, shows the multi-omics analysis (16S rRNA, metagenomics, metatranscriptomics and metabolomics) of capybara gut microbiome in cecum and recto. The authors describe the microbial composition and enzymatic systems CAZymes and PULs from Bacteroidetes along the metabolic pathways involved in the conversion of dietary cellulose to SCFA.

General comments: The manuscript is well written, and the authors demonstrate especially the importance of CAZymes families from Bacteroidetes in the gut microbiome of capybara rodent. These novel findings along the complete description of the microbial composition are of interest for the scientific community. However, I have major concerns about the research.

Major comments:

1. I did a research about the state of the art about the capybara gut microbiome, and according to it there are few works already that have describe the bacterial diversity of this specie (DOI: 10.1007/s00248-011-9963-z, 10.1128/AEM.01864-20). However, none of these works have been referenced or discussed, what are the novel findings in relation to these studies as it seems that the capybara gut microbial composition has been already studied?
2. In relation with my previous comment, there is a version of the current manuscript in a pre-print repository. Does it mean that authors have sent the same work for two journals?
3. I am concern about the sample size and that the manuscript seems very descriptive, always comparing to what has been found but not with experiments. How accurate are both issues compared to the reality? It would be good to add a line in the text to justify the number of samples and their comparison with published data.

Minor comments:

1. I suggest adding an experimental design that illustrate the purpose of the research, to make it easier to follow.
2. Line 87, please define CAZy families as it is the first time it appears in the text.
3. Line 98, define the abbreviations MT, MG, 16S
4. Line 106-108. Why the ratio MT/MG is commented as it was not statistically significant?, can this finding be supported by the sentences in lines 109-112?
5. Line 111. Fig 4? Figures should be commented in the text in order starting from 1.
6. Suppl. Table 1. Please add a column with the classification as high quality and medium quality

7. Line 119, the 24 MAGs are those of high quality? It would be good to add them in a separate table or in an extra-column of Suppl. Table 1.
8. Line 148. These GHs CAZymes are the most abundant according to the MG, so please add this information.
9. Line 159, GH5_25 and _37 do not appear in Fig 2.
10. Line 195. Please define CCs
11. Lines 204-206. I do not agree that this is the metabolomic fingerprint of the capybara gut microbiome. It should be stated that the metabolites were obtained from the polar part after a Folch extraction. So, the metabolic analysis is only done in one part of the microbiome metabolism.
12. I suggest that for all the figures in the main text and in supplementary, to add in all the number of samples, replicates, and description of all abbreviations in the figure so that the reader without seen the text could understand the picture. For example, add in the text what does KM and Kcat mean.
13. Same as before, this should be done the same for the tables, explaining all the abbreviations in a foot note. For example, what does the e-value means and how was calculated?
14. Line 355. Use the abbreviations MG and MT
15. Line 516. If all the supernatant was transferred and evaporated, this means that volume was not controlled between samples and the analytical response for each sample was different and can not be comparable.
16. Line 630-638, include in a repository the data from metabolomics analysis
17. Figure 4. It has not been described how metabolomic reconstruction was performed, please give all the details.

The aim of the study is to understand the enzymatic and metabolic pathways employed by the gut microbiota facilitating the breakdown and utilization of recalcitrant dietary polysaccharides. The study animals are monogastric herbivorous capybara, famous for their ability to depolymerize and utilize lignocellulosic biomass through microbial symbiotic mechanisms. The study is based on samples obtained from three euthanized females that were euthanized as a measure of management of Rocky Mountain Spotted Fever (RMSF) hosts. The microbiome (16S rRNA sequencing, V4 region), the metagenome and the metatranscriptome of samples from cecum and recto were analyzed, and combined with carbohydrate enzymology and X-ray crystallography to elucidate how capybara are able to convert the hard-to digest dietary fibers into short-chain fatty acids to cover their energetic demands. Overall, these is a highly interesting paper that would fit very well into the journal.

I have one major concern, and this relies on the fact that the three study individuals might have been infected with *Rickettsia*. You might argue most wildlife is infected with something – I agree. However, usually sample sizes are much higher.

The gut microbiome takes up many of these tasks and constant direct and indirect molecular crosstalk between the genomes of interacting hosts and symbionts maintain a stable gut microbial community, optimise their functionality and buffer against disturbances. Radical changes in the commensal microbial community are, however, seen in connection with parasitic infections (e.g., viruses (Wasimuddin et al., 2018 ISME, 2019 SciRep); bacteria (Borewicz et al., 2015 PLoS One); helminths (Li et al., 2016 SciRep). Parasites exploit unused metabolic products, compete for space and resources with commensal bacteria, consequently, changing the host's microbiome composition and function (Zaneveld et al., 2017 NatMicrobiology).

I am not an expert in CAZymes / X-ray / structural biology, and metabolomics aspects, and will focus on the animal microbiome perspective; especially, given that the study appears to use animals that may have been infected with Rocky Mountain Spotted Fever (RMSF), a bacterial disease spread by ticks, caused by *Rickettsia rickettsia* (Proteobacteria). Yet, I agree that this is not the focus of this paper, one might cross-check how a potential infection could affect the presented results, especially given the small sample size (for a microbiome study).

Line 95, I don't think that the 16S marker used to sequence bacteria would also be used for Archaea and Fungi.

Line 100ff, Fig. 1: Only three correlations out of six potential correlations are reported. Why are all correlations with only 16S missing? Additionally, there seems to be a shift in the abundance and at least the ranking should be different. Did you test for it? Could you check for the presence of Rickettsia among the Proteobacteria?

Fig. 1: Puzzling is the large group of "unclassified" Are these chloroplasts? If yes, is the coverage sufficient enough once the chloroplasts are taken out? See Viquez et al 2020, *EvoEcol* for a discussion of the issues that may rise with microbiome sequencing of mammals with a plant-based diet. The method section is very short and superficial regarding all the "normal" descriptive information that is usually presented, including the sequencing coverage before/after quality filtering, individual range etc.

Line 133 There are comparisons lacking here.

Fig. S1: How can you obtain nine data points? You have three individuals, cecal and rectal data...should be six data points. The individual variation is quite large.

Using zOTUs is not common. Why don't you report ASVs, as usual in most recent paper. Would QIIME give you different results?

REVIEWERS' COMMENTS:

Reviewer #1 (Remarks to the Author):

In this manuscript, Cabral et al. provide a tour de force analysis of a mammalian gut microbiome, going from metagenomic analysis to X-ray crystallography. Along the way, they discover new subfamilies of carbohydrate binding modules (CBMs) and glycoside hydrolases (GHs). The manuscript is generally well written and easy to read, if not a bit wordy and in need of a little brushing up on the English. I have no major concerns with this paper, only minor comments, including:

R: Thank you for the positive comment and for taking the time to revise our work.

1. The use of the word "deconstruction" (e.g., in line 34 but recurs throughout the text) is not ideal; probably better to use "degradation" or "depolymerization."

R: Thank you for the suggestion. It was modified in the revised version of the manuscript.

2. "Exploring the genomic dark matter..." is a wonderful turn of phrase but perhaps a little much for a scientific manuscript.

R: Thank you for pointing this out. It was modified in the revised version of the manuscript.

3. Some abbreviations are used without definition (e.g., MG and MT in line 96).

R: It was modified in the revised version of the manuscript. Abbreviations were defined in the first time it appeared in the text and in all figures.

4. In lines 130-137, are these differences statistically significant?

R: This part of the text was removed after the changes in this section following the comments from other reviewers to become this section less descriptive.

5. In lines 333-335, is this shown or just inferred?

R: We have biochemically tested each enzyme comprising the cluster CC102. Given that the CapGH97 is a calcium-activated α -galactosidase, CapGH43_12 is a highly active α -L-arabinofuranosidase, and the CapCBMXX-GH10 is an endo- β -1,4-xylanase coupled with the novel CBM targeting xylan, we have inferred that they would complementarily act on heteroxylans with substitutions of α -galactosyl and α -L-arabinofuranosyl moieties as show in the current Figure 7e.

6. In line 342, I don't understand "valorization."

R: It was modified in the revised version of the manuscript. Now it reads “

“The understanding of the enzymatic and metabolic mechanisms employed by these microbial communities to obtain energy from plant fibers may unveil alternative biological systems for the conversion of these lignocellulosic agro-industrial residues into value-added products and create new opportunities for carbohydrate-based biotechnological applications.”

7. In lines 346-348, some comment on the relative timescales of microbiome shaping versus evolution would be useful here.

R: Thanks for the suggestion. We have modified the manuscript to include the following information in the discussion section.

“These semi-aquatic animals are hindgut fermenters throughout found in Pantanal wetlands and Amazon basin and, in particular, such animals dwelling the Piracicaba basin region in Brazil have incorporated sugarcane in their diet for decades, a relevant timescale for microbial adaptation and specialization⁴¹, reasoning that this microbiota has been shaped and optimized for energy extraction²¹ from this industrially relevant lignocellulosic biomass. Sugarcane is an important feedstock for Brazilian economy and other countries such as India and Thailand. Two thirds of this crop are made of lignocellulose, which currently is left in the field (straw) and burnt for energy purposes (electricity and vapor). The understanding of the enzymatic and metabolic mechanisms employed by these microbial communities to obtain energy from plant fibers may unveil alternative biological systems for the conversion of these lignocellulosic agro-industrial residues into value-added products and create new opportunities for carbohydrate-based biotechnological applications.”

Reviewer #2 (Remarks to the Author):

Cabral and colleagues elucidated the microbial community composition of capybara gut, the largest living rodent that processes lignocellulosic biomass. The authors also unveiled the enzymatic systems and metabolic pathways involved in the conversion of recalcitrant dietary fibers into short-chain fatty acids using a multidisciplinary approach, including multi-meta-omics and enzymology. The work also provides the identification and structural characterization of two novel CAZy families: a glycosyl hydrolase (GH) and a carbohydrate binding module (CBM). Because of its novelty and importance, I recommend publishing this manuscript in Nature Communications.

R: We do appreciate the positive comment and for taking the time to revise our manuscript.

Please, find below some comments and suggestions:

Major:

1. Please, explain in more detail what are the main differences between the new GH family from the

distantly related GH5 and GH30 families using sequence alignments, secondary structure comparisons and enzymatic activities.

R: Thank you for the suggestion. These analyses were included in the revised version of the manuscript (please see below) and the Supplementary Fig. 9-11 and Supplementary Table 9 were included to support these analyses.

“Protein modelling and threading performed using RoseTTAFold³⁷ and PDBsum³⁸, respectively, revealed that CapGHXXX consists of a $(\alpha/\beta)_8$ -barrel structure (Supplementary Fig. 8), which is an archetypal scaffold of the clan GH-A. According to structural predictions, CapGHXXX exhibits a two-domain architecture including an appended β -sandwich domain (Supplementary Fig. 9), which is a similar structural organization found in the GH30 family. With the exception of the residues defining the clan GH-A, sequence alignment with GH5 and GH30 members revealed a very low sequence conservation below the criterium for significant similarity detection (using an e-value < 0.05), demonstrating that although the domains in the tertiary structure can be similar, the sequences between these families are remarkably diverse (Supplementary Fig. 9-11 and Supplementary Table 9). To further explore the GHXXX family, the enzyme BXY_26070 (SEQ_ID CBK67650.1) from *B. xylanisolvens*, which shares 46% sequence identity with CapGHXXX, was also heterologously expressed, purified and biochemically characterized (Table 1). The two members characterized from the GHXXX family present β -galactosidase activity, which is not described in either GH30 or GH5 families, strengthening at biochemical level the establishment of this new GH family.”

Current Supplementary Fig. 9: Structural comparison between CapGHXXX model and (a) GH5 and (b) GH30 members. GH5 members are the endo- β -1,4-mannanases from *Streptomyces thermolilacinus* NBRC14274 (StMan, subfamily GH5_8, PDB code 3WSU, rmsd 3.27 Å, in cyan⁹¹) and from *Streptomyces* sp. SirexAA-E (SACTE_2347, subfamily GH5_8, PDB code 4FK9, 4.41 Å rmsd, in green⁹²). GH30 members are the endo- β -1,4-xylanase from *Ruminiclostridium papyrosolvens* C71 (CpXyn30A, subfamily GH30_8, PDB code 4FMV, rmsd 2.07 Å, in blue⁹³) and the glucuronoarabinoxylan-specific endo- β -1,4-xylanase from *Erwinia chrysanthemi* (XynA, subfamily GH30_8, PDB code 1NOF, rmsd 2.12 Å, in salmon⁹⁴). **c-d**, Identification of conserved clan GH-A residues in the CapGHXXX model. In panel (c), CapGHXXX model is in orange and the GH5 endo- β -1,4-mannanases StMan (PDB code 3WSU) and SACTE_2347 (PDB code 4FK9) are in cyan and green, respectively. In panel (d), CapGHXXX model is also in orange and the GH30

CpXyn30A and XynA are in blue and salmon, respectively. The residues E305 and E207 correspond to the nucleophile and acid/base, respectively (inferred by structural superposition). Structural alignment was performed using the "super" command in Pymol (The PyMOL Molecular Graphics System, Schrödinger, LLC, New York).

Current Supplementary Fig. 10: Multiple sequence alignment between GHXX members (CapGHXXX and CBK67650 (BXY_26070)) and the GH5 members, described in Supplementary Table 9. Red triangles indicate the conserved residues from Clan GH-A and the inferred catalytic residues. ClustalΩ from MPI Bioinformatics⁹⁵ was used to generate the multiple sequence alignments, with manual adjustment based on conserved residues according to structural comparisons. ESPrpt 3.0⁹⁶ was used to generate the alignment image.

Current Supplementary Fig. 11: Multiple sequence alignment between GHXXX members (CapGHXXX and CBK67650 (BXY_26070)) and the GH30 members, described in Supplementary Table 9. Red triangles indicate the conserved residues from Clan GH-A and the inferred catalytic residues. The orange line represents the β -sandwich domain found in GH30 members. Clustal Ω from MPI Bioinformatics⁹⁵ was used to generate the multiple sequence alignments, with manual adjustment based on conserved residues according to structural comparisons. ESPrnt 3.0⁹⁶ was used to generate the alignment image.

Current Supplementary Table 9: Structural comparison of structurally characterized GH5 and GH30 members and CapGHXXX modelled structure.

Gene ID	Family_Subfamily	Activity	PDB code	Rmsd (Å)*	Identity (%) (Coverage %)
AEY82463.1	GH30_8	Glucuronoarabinoxylan-specific endo-β-1,4-xylanase	4QAW	2.06	NS
EGD48159.1	GH30_8	Endo-β-1,4-xylanase	4FMV	2.08	NS
AAA75477.1	GH5_1	Endo-β-1,4-glucanase	1VRX	2.09	NS
AAB53151.1	GH30_8	Glucuronoarabinoxylan-specific endo-β-1,4-xylanase	1NOF	2.12	21% (24%)
CAA97612.1	GH30_8	Glucuronoarabinoxylan-specific endo-β-1,4-xylanase	3GTN	2.17	NS
AAO78418.1	GH30_3	Endo-β-1,6-glucanase	5NGK	2.75	26% (11%)
AAK76864.1	GH30_8	Endo-β-1,4-xylanase	5CXP	3.23	NS
BAK26781.1	GH5_8	Endo-β-1,4-mannanase	3WSU	3.27	NS
AND74761.1	GH5_2	Endo-β-1,4-glucanase	5I2U	3.55	NS
AEN10237.1	GH5_8	Endo-β-1,4-mannanase	4FK9	4.41	26% (20%)
AAC19169.1	GH5_2	Endo-β-1,4-glucanase	1A3H	4.54	NS
AGA35556.1	GH5_36	Endo-β-1,4-mannanase	3W0K	5.49	NS
BAB04322.1	GH5_4	Endo-β-1,4-glucanase	4V2X	5.88	NS

*Structural alignment was performed using the "super" command in Pymol.

NS: No significant similarity found using Blast search with e-value < 1e-5.

2. Please, expand the overall discussion further – e.g. relevance of the findings to the industrial application of sugarcane processing? This application is mentioned in the introduction and just briefly (one sentence) in the discussion.

R: We have modified the manuscript to expand this aspect in the results and discussion sections (please see below).

In the result section: “As presented in the taxonomic analysis, Bacteroidaceae bacterium MAG57 represents a novel genome with a remarkable number of CAZyme-encoding genes including a gene cluster targeting arabinoxylan (CC102), an abundant hemicellulose in secondary cell walls of sugarcane

and other grasses. This cluster encodes two exo-enzymes from families GH43 and GH97, and an unconventional GH10 member with an unknown 45 kDa N-terminal domain (Fig. 7a).”

In the discussion section: “These semi-aquatic animals are hindgut fermenters throughout found in Pantanal wetlands and Amazon basin and, in particular, such animals dwelling the Piracicaba basin region in Brazil have incorporated sugarcane in their diet for decades, a relevant timescale for microbial adaptation and specialization⁴¹, reasoning that this microbiota has been shaped and optimized for energy extraction²¹ from this industrially relevant lignocellulosic biomass. Sugarcane is an important feedstock for Brazilian economy and other countries such as India and Thailand. Two thirds of this crop are made of lignocellulose, which currently is left in the field (straw) and burnt for energy purposes (electricity and vapor). The understanding of the enzymatic and metabolic mechanisms employed by these microbial communities to obtain energy from plant fibers may unveil alternative biological systems for the conversion of these lignocellulosic agro-industrial residues into value-added products and create new opportunities for carbohydrate-based biotechnological applications.”

Minor:

1. In the abstract the sentence “combination of unique enzymatic mechanism from Fibrobacteres to degrade cellulose with a broad arsenal of CAZymes organized in PULs from Bacteroidetes” is confusing. Please, re-write.

R: Thank you for the suggestion. The abstract was modified accordingly (please see below).

“In this microbiota, the unconventional enzymatic machinery from Fibrobacteres seems to drive cellulose degradation, whereas a diverse set of Carbohydrate-Active enZymes (CAZymes) from Bacteroidetes organized in polysaccharide utilization loci (PULs) are accounted to tackle complex hemicelluloses typically found in gramineous and aquatic plants.”

2. On page 4, line 86: define CAZY for the first time and not on page 6 line 142.

R: Abbreviations were defined in the first time it appeared in the text and in all figures.

3. Figure 4 is first introduced after Figure 1 on page 4, line 109 and on page 5, line 127. The authors could have this discussion later when Figure 4 is presented.

R: We have modified the manuscript accordingly.

4. On page 5, line 116: specify the extended form for GTDB.

R: This information was included in the revised version of the manuscript.

5. The authors suggest that the cellulose degradation in the capybara gut might be mainly accomplished by endo-beta-1,4-glucanases (EC 3.2.1.4) from families GH5, GH8, GH9 and GH45. However, the abundance of these GH families is quite low (Figure 2). Is this a common feature in other microbiota animals that eat grass or it is only found in capybara?

R: Although the endo-beta-1,4-glucanases (EC 3.2.1.4) from families GH5, GH8, GH9 and GH45 are not the most abundant CAZymes in the whole metagenome, considering the recovered MAGs these families are highly expressed in the Fibrobacteres MAGs (MAG 70-72) as shown in the Supplementary Fig. 2. In addition, it is worth to note that the relative abundance of these cellulose-degrading families is quite variable even in ruminants according to the literature. For instance, these enzymes represent approximately 2% in the metagenome of Angus cattle (Brulc et al., 10.1073/pnas.0806191105), 18% in the metagenome of Guernsey cattle (Hess et al., 10.1126/science.1200387) and 35% in the metatranscriptome of Holstein cattle, bovine (Dai et al., 10.1038/srep14567). A detailed revision can be found in (Terry et. al, 10.1139/cjas-2019-0024). In our data, these enzymes represent ~2.8% of the total identified CAZymes.

6. On page 7, line 179: remove “as reported in”.

R: It was removed.

7. Please, mention the software used to perform the prediction of the GHXXX family structural topology (Suppl. Fig. S7 and page 9, line 246).

R: During the paper evaluation period, we updated the structural model of the GHXXX using RoseTTAFold, an improved deep learning-based modeling method (Baek et al., 2021). The predictor generated a model with 0.74 of confidence, and the protein topology was obtained using PDBsum (Laskowski et al., 2018). This information and an updated version of the Supplementary Fig 7 (current Supplementary Fig 8) were included in this revised manuscript

8. Can the authors explain why the catalytic efficiency of the single GH10 domain is better than the full-length protein against rye arabinoxylan?

R: The higher catalytic efficiency of the single GH10 domain compared with the full-length protein was observed using the soluble and isolated substrate. Under such in vitro conditions, the advantage conferred by the accessory CBM domain, usually associated with targeting the catalytic domain to specific insoluble and complexed polysaccharides in the plant cell walls, is lost. Moreover, the binding-unbinding events of the CBM to the soluble substrate might represent a substrate competing condition with the catalytic domain, contributing to slowing down the catalytic rate of the full-length enzyme (Kari et al., 10.1038/s41467-021-24075-y). Another factor that may negatively affect the turnover is the reduced diffusion rates of the full-length in relation to the isolated catalytic domain, which has a smaller

gyration radius and a more compact globule-like structure. The kinetic characterization of several truncated versions of glucanases, xylanases and mannanases also verified that the accessory domain negatively affects the catalytic rates on soluble substrates (Wen et al., 10.1021/bi0500630, Marrone et al., 10.1093/protein/13.8.593, Santos et al., 10.1016/j.jsb.2011.11.021 and Santos et al., 10.1042/BJ20110869).

9. Figure 6 or the text should be reorganized. In the text, the binding experiments of CapCBMXX are described before the crystal structure and in the Figure the other way around.

R: Thanks for the suggestion. We have modified the Figure accordingly. In the revised version of the manuscript, it is Fig. 7.

10. On page 12, line 328. GH28 active site or GH10 active site?

R: It is indeed the GH28 active site. We have modified the manuscript, as shown below, to clarify this point.

“It is worth to mention that two aromatic residues considered critical for the activity of a GH28 member are present in the corresponding region of the CapGH10 β -helix domain, Y193 and Y279; however, their alanine mutation did not alter the carbohydrate binding, in agreement with the lack of catalytic function of the CapGH10 β -helix domain (Supplementary Figs. 16 and 18).”

Reviewer #3 (Remarks to the Author):

To the authors:

The original research manuscript entitled “Gut microbiome of the largest living rodent harbors unprecedented enzymatic systems to break down complex plant polysaccharides”, shows the multi-omics analysis (16S rRNA, metagenomics, metatranscriptomics and metabolomics) of capybara gut microbiome in cecum and recto. The authors describe the microbial composition and enzymatic systems CAZymes and PULs from Bacteroidetes along the metabolic pathways involved in the conversion of dietary cellulose to SCFA.

General comments: The manuscript is well written, and the authors demonstrate especially the importance of CAZymes families from Bacteroidetes in the gut microbiome of capybara rodent. These novel findings along the complete description of the microbial composition are of interest for the scientific community. However, I have major concerns about the research.

R: We would like to thank you for your positive comment and for taking the time to revise our manuscript.

Major comments:

1. I did a research about the state of the art about the capybara gut microbiome, and according to it there are few works already that have describe the bacterial diversity of this specie (DOI: 10.1007/s00248-011-9963-z, 10.1128/AEM.01864-20). However, none of these works have been referenced or discussed, what are the novel findings in relation to these studies as it seems that the capybara gut microbial composition has been already studied?

R: We have modified the manuscript to include comparisons with previously published manuscripts. However, it worth to mention that Garcia-Amado et al. 2012 and Milani et al. 2020 analyzed the bacteria composition only by 16S of Capybara cecal and fecal samples respectively, and they did not employ other omics such as metagenomics, metatranscriptomics and metabolomics on capybara samples that are instrumental to elucidate the microbial community strategies for plant fiber digestion (Note that Milani et al., 2020 performed metagenomics and metatranscriptomics, but on selected animals, which do not include capybara). Furthermore, Milani et al. 2020 aimed at understanding the microbiota composition differences among carnivores, herbivores and omnivores, using a large number of mammalian species, without focusing on capybara or on the discovery of novel enzymatic systems. The study by Garcia-Amado et al. 2012 employed hybridization on DNA microarray approach, which provided a preliminary view of the microbiota of capybara, as stated in the last paragraph of their article.

Our findings, beyond the microbial composition previously observed, include (i) the discovery of lignocellulose-degrading genomes from recovered MAGs representing taxonomic novelties, (ii) the elucidation of the molecular strategies employed by capybara gut microbiota to break down plant fibers (CAZyme inventory), (iii) the sugar-to-fatty acid metabolic conversion pathways, (iv) a new enzymatic system associated with heteroxylan degradation that encompasses a novel CBM family, which represents an unprecedented fold among the CBM superfamily, and, ultimately, (v) the founding of a novel GH family exhibiting beta-galactosidase activity. These findings highlight the great potential of this microbiota as source of enzymes for the processing of plant polysaccharides, expanding our current understanding about gut microbial strategies to overcome the recalcitrance of lignocellulosic biomass, which might be utilized in biorefineries for aggregating value to agro-industrial residues.

2. In relation with my previous comment, there is a version of the current manuscript in a pre-print repository. Does it mean that authors have sent the same work for two journals?

R: We would like to highlight that the manuscript was not concomitantly submitted to two journals. The study was deposited in a pre-print repository in accordance with journals policies and standards.

3. I am concern about the sample size and that the manuscript seems very descriptive, always comparing to what has been found but not with experiments. How accurate are both issues compared to the reality? It would be good to add a line in the text to justify the number of samples and their comparison with published data.

R: We have used fresh samples collected directly from the cecum and recto from three wild animals, which were euthanized as a strategy to control the proliferation of Rocky Mountain Spotted Fever hosts. For the collection of fresh cecal samples, an abdominal surgical procedure had to be conducted by an

authorized vet in each animal, which restricted us from using a larger number of animals. For instance, a similar sample size was employed in Solden et al 2018 Nat Microbiology (10.1038/s41564-018-0225-4). It was mentioned in the methods section.

Regarding the comparison with published data, the enzymatic and structural data were extensively compared with literature as exemplified by the several supplementary figures and tables, and many cited references. Indeed, the founding of the two novel families were only possible after detailed phylogenetic, biochemical and structural comparisons with published data. About the omics data, please see the comment #1, we have included the comparisons with the two published experiments and discussed about the relevance of our findings in the context of lignocellulose degradation. Moreover, the taxonomic section was significantly modified to focus on relevant comparisons for our study, without extending about general descriptions of the microbiota.

Minor comments:

1. I suggest adding an experimental design that illustrate the purpose of the research, to make it easier to follow.

R: Thank you for the suggestion. We have included an experimental design as bellow.

Current Fig 1. Experimental design employed to explore capybara gut microbiome. An integrated multi-omics approach from the community to the molecular level, employing 16S rRNA gene targeting sequencing (16S), metagenomics, metatranscriptomics and NMR-based metabolomics along with phylogenetics, enzymatic assays and X-ray crystallography were used to investigate the plant fiber

depolymerization strategies and major energy conversion pathways exploited by Capybara gut microbiota.

2. Line 87, please define CAZy families as it is the first time it appears in the text.

R: It was included.

3. Line 98, define the abbreviations MT, MG, 16S

R: These terms were defined in the first time they appear.

4. Line 106-108. Why the ratio MT/MG is commented as it was not statistically significant? can this finding be supported by the sentences in lines 109-112?

R: In the new version of the manuscript, this sentence was removed.

5. Line 111. Fig 4? Figures should be commented in the text in order starting from 1.

R: We have modified the manuscript properly.

6. Suppl. Table 1. Please add a column with the classification as high quality and medium quality

R: We have modified the Supplementary Table 1 to include this information.

7. Line 119, the 24 MAGs are those of high quality? It would be good to add them in a separate table or in an extra-column of Suppl. Table 1.

R: Thanks for the suggestion. The 24 MAGs are considered taxonomic novelties according to the GTDB classification. We have modified the Supplementary Table 1 and included an additional panel in the current Fig. 2c to highlight the taxonomic novelties genomes recovered from capybara gut.

8. Line 148. These GHs CAZymes are the most abundant according to the MG, so please add this information.

R: This information was included in the manuscript.

9. Line 159, GH5_25 and _37 do not appear in Fig 2.

R: The GH5 from subfamilies GH5_25 and GH5_37 was not included in the current Fig 3 since they were found in low abundance. Therefore, we found more reasonable to remove this information from the text.

10. Line 195. Please define CCs

R: It was defined in the first time it appeared in the text and in all figures.

11. Lines 204-206. I do not agree that this is the metabolomic fingerprint of the capybara gut microbiome. It should be stated that the metabolites were obtained from the polar part after a Folch extraction. So, the metabolic analysis is only done in one part of the microbiome metabolism.

R: Thank you for pointing this out. We have modified the manuscript accordingly.

12. I suggest that for all the figures in the main text and in supplementary, to add in all the number of samples, replicates, and description of all abbreviations in the figure so that the reader without seen the text could understand the picture. For example, add in the text what does KM and Kcat mean.

R: We have revised the figures to include the number of samples, replicates, and abbreviations. The meaning of kinetic parameters (Km and Kcat) were included in the text, table 1 and in the current figure 7.

13. Same as before, this should be done the same for the tables, explaining all the abbreviations in a foot note. For example, what does the e-value means and how was calculated?

R: We have revised the tables to include the meaning of the abbreviations.

14. Line 355. Use the abbreviations MG and MT

R: The manuscript was modified accordingly.

15. Line 516. If all the supernatant was transferred and evaporated, this means that volume was not controlled between samples and the analytical response for each sample was different and can not be comparable.

R: We have modified the manuscript to include a more detailed description of metabolites quantification.

“The metabolites were processed and quantified using the NMR Suite software version 7.5 (Chenomx Inc™, Edmonton, AB, Canada). Processor module of this software was used to adjust the spectral phase and baseline corrections. A 0.5 Hz line-broadening function was used to reduce signal noise and facilitate the fitting of the metabolite signals in spectral peaks. The water signal was suppressed, and the spectra were calibrated using the reference signal of the TMSP-d₄ as 0.5 mM. The spectra were individually transferred to the Profiling module of this software to determine the metabolomic profile of each group.

Metabolites were identified and their concentrations were measured. Metabolite concentration data were normalized using the initial amount of extraction.”

16. Line 630-638, include in a repository the data from metabolomics analysis

R: The metabolomics data was deposited in the Metabolomics Workbench under study ID ST001945. This information was included in the manuscript.

17. Figure 4. It has not been described how metabolomic reconstruction was performed, please give all the details.

R: The following text was included in the legend of current fig. 5 (previously fig. 4):

“To perform the metabolic reconstruction of each MAG, the annotation obtained from KOFAM database were filtered to keep only the top 5 hits of each protein (E-value < 1e-5). These filtered annotations were supplied to the Annotation of Metabolite Origins (AMON) tool, which predicts the metabolites that each MAG can produce, according to KEGG Orthologous (KO) assignments.”

Reviewer #4

The aim of the study is to understand the enzymatic and metabolic pathways employed by the gut microbiota facilitating the breakdown and utilization of recalcitrant dietary polysaccharides. The study animals are monogastric herbivorous capybara, famous for their ability to depolymerize and utilize lignocellulosic biomass through microbial symbiotic mechanisms. The study is based on samples obtained from three euthanized females that were euthanized as a measure of management of Rocky Mountain Spotted Fever (RMSF) hosts. The microbiome (16S rRNA sequencing, V4 region), the metagenome and the metatranscriptome of samples from cecum and recto were analyzed, and combined with carbohydrate enzymology and X-ray crystallography to elucidate how capybara are able to convert the hard-to digest dietary fibers into short-chain fatty acids to cover their energetic demands. Overall, these is a highly interesting paper that would fit very well into the journal.

R: Thank you for your positive comment and for taking the time to revise our manuscript.

I have one major concern, and this relies on the fact that the three study individuals might have been infected with Rickettsia. You might argue most wildlife is infected with something – I agree. However, usually sample sizes are much higher. The gut microbiome takes up many of these tasks and constant direct and indirect molecular crosstalk between the genomes of interacting hosts and symbionts maintain a stable gut microbial community, optimize their functionality and buffer against disturbances. Radical changes in the commensal microbial community are, however, seen in connection with parasitic

infections (e.g., viruses (Wasimuddin et al., 2018 ISME, 2019 SciRep); bacteria (Borewicz et al., 2015 PLoS One); helminths (Li et al., 2016 SciRep). Parasites exploit unused metabolic products, compete for space and resources with commensal bacteria, consequently, changing the host's microbiome composition and function (Zaneveld et al., 2017 Nat Microbiology). I am not an expert in CAZymes / X-ray / structural biology, and metabolomics aspects, and will focus on the animal microbiome perspective; especially, given that the study appears to use animals that may have been infected with Rocky Mountain Spotted Fever (RMSF), a bacterial disease spread by ticks, caused by *Rickettsia rickettsia* (Proteobacteria). Yet, I agree that this is not the focus of this paper, one might cross-check how a potential infection could affect the presented results, especially given the small sample size (for a microbiome study).

*R: Thank you for pointing this out. We would like to highlight that the three animals studied here were not infected with *Rickettsia rickettsii*. The sample collection for our study was performed during one of the campaigns conducted by Nunes et al., 2020 ([10.1590/0103-8478cr2020053](https://doi.org/10.1590/0103-8478cr2020053)), which focused on the reproductive control of capybaras at risk of transmission of Rocky Mountain Spotted Fever (RMSF). Briefly, animals were captured, and blood samples were collected and the serum was tested for the presence of anti-*R. rickettsii* antibodies by immunofluorescence assays. The seropositive animals were considered immune for developing rickettsemia, then underwent surgical procedures of tubal ligations in non-gestational females and vasectomy in males, to impair the reproduction of these animals. On the other hand, seronegative animals were euthanized as a strategy to control the proliferation of the bacteria *R. rickettsia*, a practice that is guaranteed by Brazilian Joint Resolution SMA/SES No. 1, of July 2016. Since we aimed to collect fresh samples directly from the cecum and recto, we only collected samples from the seronegative animals for *R. rickettsii*, which were euthanized. Furthermore, in our omics analyses we did not detect any *R. rickettsii* reads either in 16S or MT data by mapping the reads to *R. rickettsii* genome (GCF_000283955) and by taxonomic analysis. The methods section (procedures for sample collection) was expanded to indicate that samples were only collected from seronegative animals against *R. rickettsii* and that an abdominal surgical procedure was employed to collect fresh samples from cecum and recto.*

Line 95, I don't think that the 16S marker used to sequence bacteria would also be used for Archaea and Fungi.

R: We would like to highlight that the 16S rRNA marker gene was used for taxonomic analysis of Bacteria and Archaea. For this, we used the 515F-806R bacterial/archaeal primer pair, traditionally used by the Earth Microbiome Project (EMP; <http://www.earthmicrobiome.org/emp-standard-protocols/16s/>), which includes degenerated bases in both the forward and reverse primers to remove known biases against Crenarchaeota/Thaumarchaeota (515F - Parada et al., 2016) and the Alphaproteobacterial clade SAR11 (806R - Apprill et al., 2015). For Fungi taxonomy analysis, we used both metagenomics and metatranscriptomics data.

Line 100ff, Fig. 1: Only three correlations out of six potential correlations are reported. Why are all correlations with only 16S missing? Additionally, there seems to be a shift in the abundance and at least

the ranking should be different. Did you test for it? Could you check for the presence of Rickettsia among the Proteobacteria?

*R: We have modified the manuscript (methods section) to mention that a significant Pearson correlation was observed among all omics data ($P < 0.05$), except for 16S. We have further performed Kendall's rank correlation test, which resulted in a significant correlation among all omics ($P < 0.05$), except between 16S from MG and 16S datasets. The sampled animals were not infected with *R. rickettsii* (see the comment above).*

Fig. 1: Puzzling is the large group of "unclassified" Are these chloroplasts? If yes, is the coverage sufficient enough once the chloroplasts are taken out? See Viquez et al 2020, *EvoLEcol* for a discussion of the issues that may rise with microbiome sequencing of mammals with a plant-based diet. The method section is very short and superficial regarding all the "normal" descriptive information that is usually presented, including the sequencing coverage before/after quality filtering, individual range etc.

R: Thank you for the suggestion. We have found a small number of chloroplasts reads among MG dataset (0.02 – 0.1% of total number of reads) and did not find any chloroplasts reads in the MT dataset, by mapping reads with RefSeq organelles database, downloaded on October 27th, 2021. We have included this information in the additional Supplementary Table 14 along with other information regarding the number sequenced reads before and after quality filtering. For taxonomic classification, MG and MT reads were classified with Kaiju using the NCBI NR database non-redundant protein database including Bacteria, Archaea, Viruses, Fungi and microbial eukaryotes. Even for rumen and human gut microbiome that are far better studied than the capybara gut microbiome, the rate of taxonomic classification of reads was improved only by using information of a huge number of MAGs. Stewart et al 2019 (10.1038/s41587-019-0202-3) employed more than 4,900 MAGs and Passolli et al 2019 (10.1016/j.cell.2019.01.001) 150,000 MAGs. Thus, the high rate of unclassified reads suggests that there are many species in the capybara gut that are not present in the current databases.

Line 133 There are comparisons lacking here.

R: The taxonomic section was expressively modified to include more comparisons and to reduce general descriptions of the microbial composition. In the revised version, we focused on the comparisons with the previous studies from capybara gut microbiota composition and its potential correlations with feed specialization/adaptation, followed by a discussion regarding the taxonomic novelties that are involved in plant fiber degradation. The fact that these novel MAGs were further biochemically explored to reveal the two novel CAZy families, was made more explicit in this section to better connect this section with further findings. Please see the novel section for more details.

Fig. S1: How can you obtain nine data points? You have three individuals, cecal and rectal data...should be six data points. The individual variation is quite large.

R: For the 16S rRNA analysis, we have performed the reactions in technical triplicates. We have 18 data points from three individuals (cecal and rectal) and technical triplicates; however, some data points are

overlapping in the Supplementary Fig. 1. This information was included in the revised version of the manuscript.

Using zOTUs is not common. Why don't you report ASVs, as usual in most recent paper. Would QIIME give you different results

R: We have used the UPARSE-UNOISE3 algorithm for the 16S rRNA gene maker analysis, which is also an amplicon sequencing error-correction method aiming to infer accurate biological sequences from reads. The zero-radius OTUs reported by UPARSE-UNOISE3 is equivalent to Amplicon Sequence Variants (ASVs) reported by DADA2, which is also available from QIIME. Therefore, we expect to get similar results, but not necessarily the exact same one, especially regarding the number of ASVs detected by the two methods, as already shown by Prodan et al. 2020 (10.1371/journal.pone.0227434) and Nearing et al. 2018 (10.7717/peerj.5364).

REVIEWERS' COMMENTS

Reviewer #2 (Remarks to the Author):

The authors have answered all my comments/suggestions. I support the publication of the manuscript.

Reviewer #3 (Remarks to the Author):

To the authors:

Thank you for answering satisfactorily the reviewer's comments. I consider the manuscript has improved significantly and it is clearer to follow for the reader. The manuscript is of high interest to the scientific community especially for researchers in the area of sustainable carbohydrate-based technologies and those interested in knowing how to study microbiota. I just have a few minor comments:

Minor comments:

1. Line 73. Please define NMR
2. Section at line 121. Please add if the studies in this section were carried out using cecum or colon samples or both.
3. Figure 5 caption, line 921. E value is with uppercase but in the text was with lower case, please check.
4. Figure 7 caption, line 936. Define WT, and in line 937, the word control is repeated twice

Reviewer #4 (Remarks to the Author):

I appreciate the work of the authors in revising the MS!

All my concerns have been addressed.

REVIEWERS' COMMENTS

Reviewer #2 (Remarks to the Author):

The authors have answered all my comments/suggestions. I support the publication of the manuscript.
Thank you for the careful revision of our manuscript.

Reviewer #3 (Remarks to the Author):

To the authors:

Thank you for answering satisfactorily the reviewer's comments. I consider the manuscript has improved significantly and it is clearer to follow for the reader. The manuscript is of high interest to the scientific community especially for researchers in the area of sustainable carbohydrate-based technologies and those interested in knowing how to study microbiota. I just have a few minor comments:
Thank you for the careful revision of our manuscript.

Minor comments:

1. Line 73. Please define NMR

It was defined.

2. Section at line 121. Please add if the studies in this section were carried out using cecum or colon samples or both.

Studies were conducted with both cecal and rectal samples. This information was included in the manuscript.

3. Figure 5 caption, line 921. E value is with uppercase but in the text was with lower case, please check.
It was modified accordingly.

4. Figure 7 caption, line 936. Define WT, and in line 937, the word control is repeated twice
It was modified.

Reviewer #4 (Remarks to the Author):

I appreciate the work of the authors in revising the MS!

All my concerns have been addressed.

Thank you for the careful revision of our manuscript.